# Uncertainties in the effects of organic aerosol coatings on polycyclic aromatic hydrocarbon concentrations and their estimated health effects

Sijia Lou[1,2], Manish Shrivastava[3], Alexandre Albinet[4], Sophie Tomaz[4,5], Deepchandra Srivastava[4,6], Olivier Favez[4], Huizhong Shen[7], Aijun Ding[1,2]

[1]Joint International Research Laboratory of Atmospheric and Earth System Sciences, School of Atmospheric Sciences, Jiangsu Provincial Collaborative Innovation Center of Climate Change, Nanjing University, Nanjing, 210023, China

[2]Frontiers Science Center for Critical Earth Material Cycling, Nanjing University, Nanjing 210023, China

[3]Pacific Northwest National Laboratory, Richland, WA 99354, USA

[4]Institut Nationala de l'Environnement industriel et des RISques (INERIS), 60550, Verneuil en Halatte, France

[5]now at : INRS, 1 rue du Morvan CS 60027, 54519, Vandoeuvre-lès-Nancy, France

[6]now at: School of Geography Earth and Environmental Science, University of Birmingham, Edgbaston, Birmingham, UK B15 2TT

[7]Shenzhen Key Laboratory of Precision Measurement and Early Warning Technology for Urban Environmental Health Risks, School of Environmental Science and Engineering, Southern University of Science and Technology, Shenzhen 518055, China

*Correspondence to*: Sijia Lou (lousijia@nju.edu.com); Manish Shrivastava (ManishKumar.Shrivastava@pnnl.gov)

**Abstract.**

We used the CAM5 model to examine how different particle-bound Polycyclic Aromatic Hydrocarbon (PAH) degradation approaches affect the spatial distribution of benzo(a)pyrene (BaP). Three approaches were evaluated: NOA (no effect of OA coatings state on BaP), Shielded (viscous OA coatings shield BaP from oxidation under cool and dry conditions), and ROI-T (viscous OA coatings slow BaP oxidation in response to temperature and humidity). Results show that BaP concentrations vary seasonally, influenced by emissions, deposition, transport, and degradation approach, all of which are influenced by meteorological conditions. All simulations predict higher population-weighted global average (PWGA) fresh BaP concentrations during December-January-February (DJF) compared to June-July-August (JJA), due to increased emissions from household activities, and reduced removal processes during colder months. The Shielded and ROI-T approaches, which account for OA coatings, resulting in two to six times higher BaP concentrations in DJF compared to NOA. The Shielded simulation predicts the highest PWGA fresh BaP concentration (1.3 ng m$^{-3}$), with 90% of BaP protected from oxidation. In contrast, the ROI-T approach forecasts lower concentrations in mid-to-low latitudes, as it assumes less effective OA coatings under warmer, more humid conditions. Evaluations against observed BaP concentrations show the Shielded approach performs best, with a normalized mean bias (NMB) within ±20%. The combined incremental lifetime Cancer risk (ILCR) for both fresh and oxidized PAHs is similar across simulations, emphasizing the importance of considering both forms in health risk assessments. This study highlights the critical role of accurate degradation approaches in PAH modelling.

## 1 Introduction

Polycyclic aromatic hydrocarbons (PAHs), emitted from incomplete combustion of biofuels and fossil fuels, are persistent organic pollutants composed of multiple aromatic rings. Some of them are contaminants of global concern due to their well-known carcinogenic and mutagenic properties, which increase the risk to human health [Boffetta et al., 1997; Perera, 1997; Chen and Liao, 2006; IARC, 2010; Kim et al., 2013; Muir et al., 2019]. For instance, in 1976, the United States Environmental Protection Agency (US EPA) listed 16 PAHs as priority pollutants [Keith 2015]. Among these, particle-bound PAHs are more carcinogenic than gas-phase PAHs [Y Liu et al., 2017]. Therefore, benzo(a)pyrene, one of the most carcinogenic PAHs and predominantly existing in the particle phase, is often used as an indicator of cancer risk resulting from exposure to PAH mixtures [EPA 2004; EPCEU 2004; MEPPRC 2009; CPCB, 2020; IARC, 2021]. Considering that lifetime exposure to 0.1 ng m$^{-3}$ of BaP would increase the additional lung cancer risk by one in 100,000 exposed persons, the World Health Organization (WHO) recommends limiting BaP concentrations to 0.1 ng m$^{-3}$ [WHO, 2000; Bostrom et al., 2002].

High levels of BaP in ambient air have been measured globally over the past two decades, ranging from 0.1 to 2.5 ng m$^{-3}$ in Europe and North America, with even higher concentrations observed in rural areas of China and India, exceeding 10 ng m$^{-3}$ [Lee et al., 2011; W Wang et al., 2011; Kim et al., 2012; Brown et al., 2013; Hu et al., 2017; 2018; Radonić et al., 2017; Ma et al., 2018; J Han et al., 2019; Lhotka et al., 2019; Munyeza et al., 2019; Ahad et al., 2020; Kumar et al., 2020]. However, compared to measurements, previous regional or global models suffer from large uncertainties, with biases spanning several orders of magnitude, largely due to an incomplete understanding of the complex gas-particle partitioning [Friedman et al., 2014; Galarneau et al., 2014; Lammel et al., 2015; Shrivastava et al., 2017; Mu et al., 2018; F Han et al., 2022]. For example, Iakovides et al. (2021) reported that using an octanol-air partition coefficient absorption model, such as the Junge-Pankow model, the gas-particle fractions of simulated PAHs are more suitable for remote or rural areas but not for urban areas. To differentiate between aerosols in European urban or rural areas, Arp et al. (2008) developed polyparameter linear free energy relationships (ppLFER) equations. Shahpoury et al. (2016) reported that the ppLFER model can distinguish a variety of organics, including liquid water-soluble/organic soluble organics, and solid/semisolid organic polymers, as well as the inorganic phases of aerosols. Therefore, by adopting the ppLFER scheme, the gas-particle partitioning of simulated PAHs in anthropogenically impacted areas is improved, and the simulated PAHs show good agreement with observations [Tomaz et al., 2016; Kelly et al., 2021].

The lack of clarity regarding the chemical loss of PAHs is a significant factor contributing to large deviations in model-simulated BaP concentrations compared to measured values. As a semi-volatile compound, BaP in the gas-phase undergoes degradation through various pathways, primarily involving reactions with hydroxyl radicals (OH) and nitrate radicals (NO$_3$), along with photolytic processes driven by light. In a particle-bound state, while BaP can also be degraded by OH and NO$_3$, this occurs at a much slower rate compared to degradation by ozone, which serves as the primary mechanism in this phase [Keyte et al., 2013]. Laboratory studies have shown that particle-bound BaP can undergo rapid oxidation within hours through heterogeneous chemical degradation of BaP on the surface of black carbon (BC), organic carbon (OC), and sulfate aerosols [Pöschl et al., 2001; Kwamena et al., 2004; Kahan et al., 2006; Zhou et al., 2012]. Despite laboratory findings, field measurements have revealed that BaP persists in the atmosphere for extended periods and can be transported over long distances, reaching even the Arctic

[Halsall et al., 1997; Masclet et al., 2000; Schauer et al., 2003; Lohmann and Lammel, 2004; Van
Overmeiren et al., 2024]. A recent laboratory study demonstrated that the presence of secondary organic
aerosol (SOA) coatings could shield BaP from ozone oxidation [Zelenyuk et al., 2012]. Based on this,
Friedman et al. (2014) used an exponential decay function, assuming that 80% of SOA-bound PAHs
were still present after 24 hours. However, the shielding effectiveness of PAHs depends on the phase
state of SOA, which should be temperature- and relative humidity-dependent [Koop et al., 2011; Zhou
et al., 2013; Berkemeier et al., 2016; Shiraiwa et al., 2017; Shrivastava et al., 2017; Mu et al., 2018].
Shrivastava et al. (2017) developed a new PAH modeling approach in the global Community Atmosphere
Model, assuming that viscous SOA can completely inhibit particle-bound PAHs (i.e., BaP) oxidation
reactions under cool or dry conditions. Implementing this approach significantly improved the agreement
between simulated and measured BaP concentrations at hundreds of locations worldwide compared to
models that ignored the shielding effects of SOA coatings. Meanwhile, Mu et al. (2018) suggested that
shutting off particle-bound BaP degradation based on the simple thresholds of temperature and relative
humidity used in Shrivastava et al. (2017) cannot represent the complex multiphase reactions of BaP.
They proposed a new ROI-T approach, accounting for the effects of temperature and humidity on SOA
phase state and BaP degradation chemical reaction rate. The BaP concentrations simulated using the
ROI-T approach exhibited the best agreement with measurements at Xianghe (China) and Gosan (South
Korea) sites [Mu et al., 2018]. However, their simulations still showed a significant underestimation of
BaP concentrations for European and Arctic background sites.
Although simulations of PAHs have significantly improved over the past decade [Sehili et al., 2007;
Friedman & Selin, 2012; Shen et al., 2014; Shrivastava et al., 2017; Mu et al., 2018; Wu et al., 2024],
particularly in terms of lifetime estimation, understanding of the oxidation chemistry remains a key area
of development. The oxidation of particle-bound BaP is highly dependent on the concentrations of
oxidants (primarily ozone) and the effectiveness of shielding by viscous organic aerosol (OA) coatings,
which are influenced by temperature and relative humidity (RH). This dependence results in notable
seasonal variations in both fresh BaP concentrations and oxidized BaP. Since assessments of PAH-
induced lung cancer risks often rely on modeled BaP concentrations [Shen et al., 2014; Shrivastava et
al., 2017; F Han et al., 2020; 2022; Famiyeh et al., 2021; Li et al., 2022; Wu et al., 2024], uncertainties
in these simulations can have significant implications for estimates of PAH exposure and associated
human health risks. This study systematically evaluates the uncertainty in simulated BaP concentrations
due to varying chemical mechanisms of BaP oxidation, considering seasonal variations. This paper also
assesses the strengths and limitations of current PAH modelling approaches, offering insights into future
simulation improvements. The structure of the paper is organized as follows: Section 2 introduces the
model, particle-bound BaP degradation approaches, emissions, and observation data used in this study.
Section 3 first presents the simulated fresh and oxidized BaP concentrations in winter and summer,
followed by a detailed comparison between simulated BaP and measurements, as well as an assessment
of PAH-related lung cancer risk. Section 4 gives the conclusions and implications for discussions.

## 2 Methods

### 2.1 Measurements

We collected observed fresh BaP concentrations at 66 background/remote sites and 208 non-background sites worldwide (Table S1, S2). The observation data of fresh BaP were obtained from the Integrated Atmospheric Deposition Network (IADN, available from https://www.epa.gov/great-lakes-monitoring/great-lakes-integrated-atmospheric-deposition-network), the European Monitoring and Evaluation Programme (EMEP, available from https://www.emep.int [Tørseth et al., 2012]), the Global Environmental Assessment Information System (GENASIS, available from https://www.genasis.cz [Boruvkova et al., 2015]), the Arctic Monitoring and Assessment Programme (AMAP [Hung et al., 2010]), and previous studies [Shen et al., 2014; Shrivastava et al., 2017]. For oxidized BaP, measurements were available from only two locations: Grenoble—an urban site situated at 5.73°E, 45.16°N—in 2013, and SIRTA—a background site located at 2.15°E, 48.71°N (http://sirtaa.ipsl.fr/)—in the years 2014-2015. However, due to the measurement limitations, data on oxidized BaP (primarily nitro-BaP) were only available from two sites in France. Therefore, this study only includes concentrations of oxidized BaP from Grenoble (an urban site located at 5.73E, 45.16ºN) in 2013 [Tomaz et al., 2016] and from the ACTRIS SIRTA atmospheric supersite (Site Instrumental de Recherche par Télédétection Atmosphérique, which is representative of the suburban background site in the Paris region, located at 2.15°E, 48.71°N; http://sirtaa.ipsl.fr/) in 2014-2015 [Lanzafame et al., 2021]. The Grenoble site is centrally located and represents a location with significant urban influence, while the SIRTA site is located 25 km southwest of central Paris and is considered representative of regional background air quality.

### 2.2 Overview of the model

We employed the global Community Atmosphere Model version 5.2 (CAM5) to simulate the global distribution of BaP concentrations. Tracer concentrations obtained from CAM5 simulations were performed at a horizontal resolution of 1.9 º latitude by 2.5 º longitude, and a vertical resolution of 30 layers between the surface and 3.6 hPa. The Model for Ozone and Related Chemical Tracers (MOZART-4) represented the gas-phase chemical mechanism [Emmons et al., 2010], while the properties and processes of aerosol species were included in the Modal Aerosol Model (MAM3) [X Liu et al., 2012]. The model encompassed six aerosol species, including inorganic aerosols (e.g. mineral dust, black carbon, sulfate, and sea salt) and organic aerosols (primary organic aerosol and secondary organic aerosol). In addition, this study utilized an update of the volatility basis-set (VBS) approach developed by Shrivastava et al. (2015). The VBS approach tracked SOA formation based on SOA precursor gas sources, addressing both functionalization and fragmentation reactions during multi-generational aging of SOA precursor gases, as well as oligomerization reactions of SOA. The ppLFER (polyparameter linear free energy relationships) model was applied to the gas-particle partitioning of BaP, encompassing both BaP absorption into organic aerosols and adsorption onto the surface of black carbon aerosol [Shahpoury et al., 2016]. Following Shrivastava et al. (2017), we divided OA into the liquid water-soluble/organic soluble phase and the solid/semi-solid organic polymer phase. More than 90% of particle-bound BaP is absorbed within organic aerosols after applying the ppLFER model. The transport, dry deposition, and

wet removal of particle-bound BaP (including oxidized PAH) are treated similarly to other aerosol
species in CAM5 [X Liu et al., 2012].
The viscosity of OA affects the atmospheric lifecycle of BaP in two ways: (1) through the gas-particle
partitioning of SOA, which also impacts the gas-particle partitioning of BaP (as d escribed by the pp-
LFER approach), and (2) by altering the heterogeneous degradation kinetics of BaP with ozone. In the
following section, we describe the application of three different model sensitivity formulations that
account for the role of OA viscosity on the lifecycle of BaP.
**2.3 BaP degradation**
The model incorporates the gas-phase reaction of BaP with OH. Consistent with previous studies, the
second-order rate coefficient for the reaction of gaseous BaP with OH is set at $5\times10^{-11}$ cm$^3$ molecules$^{-1}$
s$^{-1}$ [Keyte et al., 2013; Shrivastava et al., 2017]. Heterogeneous reactions of particulate-phase BaP with
OH and ozone are also included in the model [Cazaunau et al., 2010; Zhou et al., 2012; 2013; Keyte et
al., 2013]. The second-order rate coefficient for the reaction of particle-bound BaP with OH is determined
to be $2.9\times10^{-13}$ cm$^3$ molecules$^{-1}$ s$^{-1}$ [Esteve et al., 2006], which is two orders of magnitude slower than
the gas-phase reaction rate of BaP with OH. Conversely, particle-bound BaP reacts rapidly with ozone
within a few hours, representing the primary oxidation pathway for BaP. Note that the photolysis of BaP
is not included in this study, partly because its photolysis rate constant is much lower compared to that
of low molecular wight PAHs [Niu et al., 2007], and the current model already underestimates BaP
concentrations.
In this study, three approaches are implemented to estimate particle-bound BaP degradation, providing
insights into the uncertainty associated with this process.
(1) In the default NOA approach, the characteristics of the organic coating - such as its thickness and
viscosity - do not affect the heterogeneous loss of particle-bound BaP. In this approach, the oxidation
of BaP follows the Langmuir-Hinselwood mechanism, meaning that the reaction rate (k) is first-
order with respect to BaP and depends on ozone concentration. Since the reference state of the
organic coating in this approach is thin and liquid-like, diffusion limitations are not considered
significant [Zhou et al., 2012; 2013]. In addition, this approach does not consider the effects of OA
viscosity on the gas-particle partitioning of SOA, as it uses the FragSVSOA treatment described in
Shrivastava et al. (2015), which assumes SOA as semi-volatile liquid-like well-mixed solution
throughout its atmospheric lifetime.
(2) Following Shrivastava et al. (2017), the SOA Shielded approach is implemented, accounting for the
shielding of BaP by viscous SOA coatings. The kinetics of the heterogeneous oxidation of BaP with
ozone become much slower after absorption by organic aerosols, as thick OA coatings reduce the
kinetics of mass transfer of BaP from the interior of the particle to the particle surface. The
effectiveness of SOA shielding is related to its thickness and viscosity, influenced by temperature
and relative humidity [Zhou et al., 2012; 2013]. In this approach, when SOA coatings are less than
20 nm, we assume that SOA cannot effectively shield particle-bound BaP, and thus, the
heterogeneous oxidation kinetics remain the same as in the default NOA approach. Thick SOA
coatings (> 20 nm) can completely turn off the particle-bound BaP heterogeneous loss kinetics under
dry or cool conditions (RH < 50% or temperature < 296 K). Different oxidation kinetics with ozone
are applied under humid (RH $\geq$ 50%) and warm (temperature $\geq$ 296 K) conditions with thick SOA

coatings, where the second-order rate coefficient for the reaction of particle-bound BaP with ozone
is 14 and $6.2 \times 10^{-15}$ $cm^3$ molecules$^{-1}$ s$^{-1}$ under moderate humidity (50%$\leq$RH<70%) and high
humidity conditions (RH$\geq$70%), respectively [Zhou et al., 2013; Shrivastava et al., 2017].
(3) Following Mu et al. (2018), the ROI-T approach is implemented, accounting for the temperature
and humidity dependence of the phase state, diffusivity, and reactivity of particulate-phase BaP.
First-order reaction rate coefficients for BaP ozonolysis are sensitive to both temperature and RH
below room temperature (296 K), but are only temperature sensitive above room temperature [Mu
et al., 2018] (Table S1). Under cool and dry conditions, the first-order reaction rate coefficients are
3-4 orders of magnitude lower than those under warm conditions (Table S1). Notably, the ROI-T
approach yields a much slower oxidation reaction of particle-bound BaP than the default NOA
approach under cool and dry conditions but a faster oxidation reaction rate under warm conditions.
Note that both the Shielded and ROI-T approaches described above consider the impact of aged semi-
solid SOA on the gas-particle partitioning of fresh SOA precursors, using the FragNVSOA treatment
described in Shrivastava et al. (2015). This treatment assumes that, once SOA is formed, it is transformed
into a highly viscous, non-volatile semi-solid within the same global model timestep (30 min) due to
particle-phase oligomerization reactions within the SOA [Shrivastava et al., 2015]. The FragNVSOA
treatment also assumes that any further gas-phase organic oxidation products that condense do not form
a solution with pre-existing OA. This assumption is supported by recent experimental studies, which
show a short aging timescale of ~20 min, during which oligomer and organosulfate formation within
isoprene SOA, leading to non-equilibrium partitioning behaviour and phase transition to semi-solid SOA
[Chen et al. 2023].
**2.4 Model Sensitivity Simulations**
We conduct simulations using CAM5 to explore the uncertainty of seasonal variations in BaP
concentrations with different PAH oxidation approaches. Three effects of OA coatings on particle-bound
PAH oxidation, as detailed in section 2.3, are considered. Hence, sensitivity simulations are performed
as follows:
(1) NOA;
(2) Shielded [Shrivastava et al., 2017];
(3) ROI-T [Mu et al., 2018].
All simulations are conducted over two years (2007-2008), with the first year allocated for spin-up. Since
most observations occurred around 2004-2009, winds and temperature are nudged toward ERA-Interim
data from January 2007 to December 2008 in this study.
**2.5 Emissions**
This study utilizes the Global Emission Modeling System (GEMS) 0.1º × 0.1º global BaP emission
inventory with temporal and spatial variations, which is available from gems.sustech.edu.cn. The
inventory was derived from the PKU-PAH Global Emission Inventory [Shen et al., 2013], which
incorporates data from all major fuel consumption sources and industrial processes. The spatiotemporal
changes in global BaP emissions are detailed in our previous work [Shrivastava et al., 2017; Lou et al.,
2023]. Anthropogenic emissions, including BC, OC, and precursor gases for both secondary aerosols
and ozone, are sourced from the HTAP_v2.2 2008 emission inventory [Janssens-Maenhout et al., 2015].
Additionally, emissions from agricultural waste burning and open biomass burning emissions are
obtained from the Emissions Database for Global Atmospheric Research (EDGAR v4.3) and Global Fire
Emissions Database (GFED3.0), respectively [van der Werf et al., 2010; Crippa et al., 2018]. To maintain
consistency with the modeling timeframe and facilitate comparison with observations, all emissions are
set at 2008 levels.
**2.6 Global model downscaling formulation**
To enhance the comparison between simulated BaP concentrations and observational data, particularly
in anthropogenically influenced areas such as urban regions, we implemented a downscaling approach
based on a Gaussian diffusion algorithm. This method refined the model-derived near-surface BaP
concentrations from the coarse resolution of 2.5°×1.9° to a finer resolution of 0.1°×0.1°, aligning with
the original resolution of emission inventory. Following the methodology outlined by Shen et al. (2014),
we assigned a weighting factor ($W_i$) to each 0.1°×0.1° receiving grid cell. This factor was determined by
summing the contributions of emissions from all 0.1°×0.1° emission grid cells within a defined
neighbourhood. The neighbourhood consists of the 2.5°×1.9° model grid cell containing the receiving
grid cell and the eight adjacent model grid cells, encompassing approximately 4275 0.1°×0.1° emission
grid cells. The formulation for $W_i$ is as follows:
$$W_i = \sum_{j=1}^{n} \frac{2.03 Q_j f_j e^{-r_d t_{ji}}}{u_j \sigma z_j x_{ji}} \qquad (1)$$
Here, $Q_j$ (ng/s) represents the emissions density of the jth emission grid cell. $f_j$ (dimensionless) and
$u_j$ (m/s) are wind frequency (0-1) and wind speed in directions 1 to 16 (N, NNE, NE, NEE, E, SEE, SE,
SSE, S, SSW, SW, NWW, NW, and NNW) in the jth grid cell, respectively, taken from the ERA-interim
reanalysis wind field. The degradation rate $r_d$ (/s) involves the gas-phase reaction with OH and the
particle-phase heterogeneous reaction with ozone in the receiving grid cell derived from the chemical
transport model. n represents the number of emission grid cells within the nine model grid cells and is
approximately 4275, though the actual value depends on the alignment of the grid resolutions.
Additionally, $t_{ji}$ (s) and $x_{ji}$ (m) denote the distance and transport time from the jth emission grid cell
to the ith receiving grid cell, and $\sigma$ (m) is the vertical standard deviation of the concentrations. Finally,
the calculated $W_i$ is used as a proxy to disaggregate the model-calculated concentration of each 2.5°×1.9°
model grid cell to a 0.1°×0.1° grid cell. Previous studies have reported substantial improvements in the
distribution and magnitude of observed BaP concentrations through this downscaling process in similar
simulations [Shen et al., 2014; Shrivastava et al., 2017; Lou et al., 2023].

**2.7 Incremental Lifetime Cancer Risk**
The Incremental Lifetime Cancer Risk (ILCR) induced by exposure to PAHs in ambient air is calculated
using the following formula [Shen et al., 2014]:
$$ILCR = CSF \times LADD \times SUS = CSF \times \frac{C \times IR \times y}{BW \times LE} \times SUS \quad (2)$$
where CSF, LADD, and SUS represent the cancer slope factor, lifetime average daily dose, and a factor
describing individual susceptibility, respectively. Following Shen et al. (2014), CSF of 26.6 kg (body
weight)·day/mg for BaP was adopted as the maximum likelihood estimate based on epidemiological data

from studies on coke oven workers, using a multistage type model [U.S. EPA, 1982]. SUS accounts for individual variations in susceptibility and is defined as the product of genetic susceptibility (GeneSus), ethnicity-adjusted factor (EAF), and age-sensitivity factor (ASF), respectively. GeneSus represents the impact of genetic variations on an individual's susceptibility to BaP-induced cancer risk. Different genotypes may lead to variations in metabolic activation or detoxification of PAHs, affecting carcinogenic risk. EAF was calculated based on the lung cancer incidences for individual ethnicities reported by the United States Cancer Statistics (available from https://www.cdc.gov/united-states-cancer-statistics/index.html), excluding the effects of smoking. ASF accounts for age-related differences in susceptibility to BaP exposure. Weighting cancer risk by a factor of 10, 2, and 1 were used for the age groups of < 2, 2-16, and > 16 years, respectively [CA EPA, 2009].

LADD is calculated from BaP exposure concentration (C, mg m$^{-3}$), which is downscaled from model-predicted BaP concentrations in this study, inhalation rate (IR, m$^3$/day), exposure duration (y, year), body weight (BW), and average life expectancy of the global population (LE, 70 years). ILCR in this study is a population-weighted average and represents the maximum likelihood estimate; the unit for ILCR is one death per 100,000 persons.

**3 Results**

**3.1 Simulation of seasonal variations in global fresh BaP**

Given that lifetime exposure to 0.1 ng m$^{-3}$ of BaP theoretically results in an additional lung cancer death per 100,000 exposed persons, the WHO recommends a limit of 0.1 ng m$^{-3}$ [Bostrom et al., 2002]. BaP degradation approaches can significantly impact BaP concentrations, further influencing the assessment of PAH exposure risks in various regions. Here, in this study, we investigated three different particle-bound BaP degradation approaches related to the OA coating hypothesis to examine their effects on the spatial distribution of BaP. Considering that the effectiveness of OA coatings is strongly dependent on temperature and humidity variations, we analyzed the distribution of BaP concentrations under different seasons.

In DJF (December-January-February), population-weighted global average (PWGA) BaP concentrations with different particle-bound BaP degradation approaches are predicted to be 0.24-1.38 ng m$^{-3}$, consistently exceeding the WHO recommendation. High levels of BaP concentrations are simulated to appear in East Asia, South Asia, North Africa, and Europe, with the peak BaP exposure in eastern China exceeding 1.0 ng m$^{-3}$ (Fig. 1). In contrast, BaP concentrations are much lower in JJA (June-July-August), with PWGA values of 0.04-0.17 ng m$^{-3}$. These results indicate that the simulated BaP exhibits strong seasonality, primarily influenced by changes in emissions, deposition, and BaP degradation chemistry. Our simulated seasonal BaP concentrations, particularly for the ROI-T approach, align with a previous study using the same emissions and particulate-BaP degradation approach [Wu et al., 2024].

In 2008, residential biomass use contributed more than 60% of total atmospheric BaP emissions for households cooking, heating, and lighting [Shen et al., 2013]. Since the demand for heating and lighting is higher in winter than in summer, more residential biomass burning is required in winter, inevitably producing BaP. In addition, less precipitation in the Northern Hemisphere in winter compared to summer (Fig. S1), linked to less efficient wet removal, contributes to the seasonal variations of BaP. Thus, without

the impacts of OA coatings on BaP degradation, the seasonal variations of BaP concentrations in
simulations using the NOA approach primarily represent the changes in emissions and deposition.

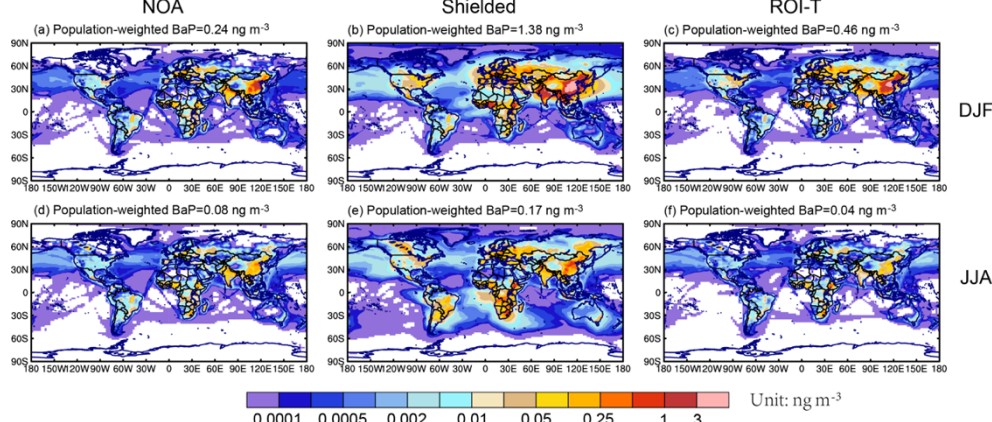


**Figure 1. The spatial distribution of simulated BaP concentrations in (a-c) DJF and (d-f) JJA. Fresh BaP**
**concentrations with different heterogeneous reaction approaches for particle-bound BaP are shown in the**
**left column (NOA), the middle column (Shieled), and the right column (ROI-T), respectively.**

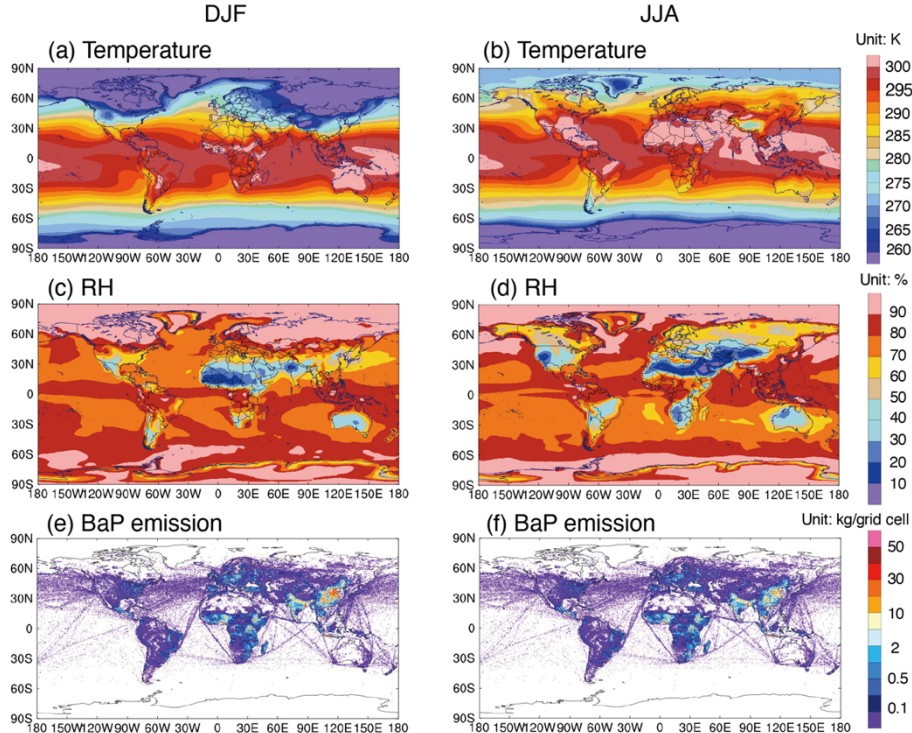


**Figure 2. The spatial distribution of surface-layer average temperature (top panel, unit: K), relative humidity**
**(middle panel, unit: %), and spatial distribution of BaP emissions (bottom panel, unit kg/grid cell) in DJF**
**(December-January-February) and JJA (June-July-August), respectively.**

Compared with simulations using the NOA approach, those incorporating OA coatings can effectively
impede the BaP loss process, leading to a significant increase in BaP concentrations during winter (Fig.
1). Upon absorption by organic aerosols, the presence of viscous OA coatings substantially hinders the
mass transfer kinetics of BaP from the particle core to the surface. The denser the organic aerosol, the
slower the diffusion of BaP, consequently slowing down the heterogeneous reactions of particle-bound
BaP with ozone and OH on the aerosol surface. This effect is more pronounced in winter than in summer,
attributed to cooler conditions that likely increase the viscosity of SOA [Shrivastava et al., 2017].
For example, the PWGA BaP from the simulation using the Shielded approach is six times higher than
that in the simulation using the NOA approach in DJF. With the Shielded approach, the OA coating is
assumed to be sufficiently tacky to prevent BaP from undergoing heterogeneous reactions with ozone
completely under dry or cool conditions, thereby extending the lifetime of BaP. During Northern
Hemisphere winters, effective OA shielding occurs in areas characterized by cool temperatures (<296 K)
or dry conditions (RH<50%), covering most of the regions with high BaP emission densities (Fig. 2a-b)
[Shiraiwa et al., 2011; Saukko et al., 2012; Zhou et al., 2012; Bateman et al., 2015]. Furthermore, BaP
with OA coatings can be transported over long distances to remote areas, including the Arctic. Treating
OA coating effectiveness as the ROI-T approach, the BaP concentrations also increase, with PWGA BaP
estimated to be twice as high as in the simulation with the NOA approach during winter. Compared with
the Shielded simulation, BaP concentrations in the ROI-T simulation exhibit similar spatial patterns in
high latitudes such as Europe, northern China, and the Arctic, but lower concentrations in southern China,
South Asia, and North Africa. The ROI-T approach assumes that the diffusion coefficients of BaP and
ozone within OA coatings decrease with reducing temperature and relative humidity, thus reducing the
degradation rate of BaP. That is, under cold (<273 K) or dry (<50%) conditions, such as mid-to-high
latitudes in winter (Fig. 2a, b), the degradation rates of BaP in the ROI-T approach are two to four orders
of magnitude smaller than those without the OA coating effect. In contrast, the OA coating in southern
China, South Asia and Africa is not as effective as those in Europe, northern China, and the Arctic,
resulting in BaP concentrations similar to NOA simulation.
In JJA, BaP concentrations tend to concentrate near the source areas. While BaP concentrations in the
simulation using the Shielded approach are estimated to be higher than those in the NOA simulation, the
concentrations found in the simulation using the ROI-T approach are even lower. The ROI-T approach
assumes that the diffusion coefficients of BaP and ozone increase with temperature, leading to an
estimated faster degradation rate of BaP than in NOA and Shielded approach simulations at conditions
above room temperature. Our results are consistent with previous studies [Shrivastava et al., 2017; Mu
et al., 2018].
**3.2 Simulation of seasonal variations in global oxidized BaP**
Previous modeling studies assumed that fresh PAHs are completely degraded after oxidation [Sehili and
Lammel, 2007; Matthias et al., 2009; Friedman et al., 2014]. However, laboratory experiments suggested
that several oxidized PAHs may remain particle-bound and even increase in molecular weight [Ringuet
et al., 2012; Zelenyuk et al., 2012; Jariyasopit et al., 2014]. Furthermore, not only fresh BaP but also
certain oxidized BaP species and derivatives exhibit toxicity [EHC 2003; Clergé et al., 2019; Hrdina et
al., 2022; Peng et al., 2023]. Therefore, it is essential to understand the impact of different BaP
degradation approaches on oxidized BaP.
In this study, we tracked oxidized particle-bound BaP, which is formed through the heterogeneous
reactions of particulate-phase BaP with OH and ozone. Figure 3 shows the spatial distribution of
simulated oxidized BaP concentrations in DJF. In the absence of an OA coating, particle-bound BaP is

always available to react with O₃. Therefore, most particle-bound BaP is rapidly oxidized near source areas, with a PWGA oxidized BaP concentration of 1.3 ng m⁻³, or 82% of the total (sum of fresh and oxidized) BaP (Fig. 3a, d). If oxidized BaP is as toxic as fresh BaP, the oxidized BaP concentration in NOA globally exceeds the WHO recommendation of 0.1 ng m⁻³ by a wide margin. In comparison, the Shielded simulation predicts that high levels of oxidized BaP only appear in the tropics in winter (Fig. 3b), because OA coatings are less effective at protecting BaP from ozone attack under high temperature and high RH conditions (Fig. 2a, b). Since the viscous OA coatings completely shut down the particle-bound BaP oxidation reaction under cool or dry conditions, most fresh BaP can stay in the atmosphere for several days, with only 10% of the total BaP being oxidized. Surprisingly, although the OA coating slowed the diffusion of particle-bound BaP from inside the interior of the OA to the particle surface in the ROI-T simulation, 71% of the total BaP was still oxidized on a global basis.

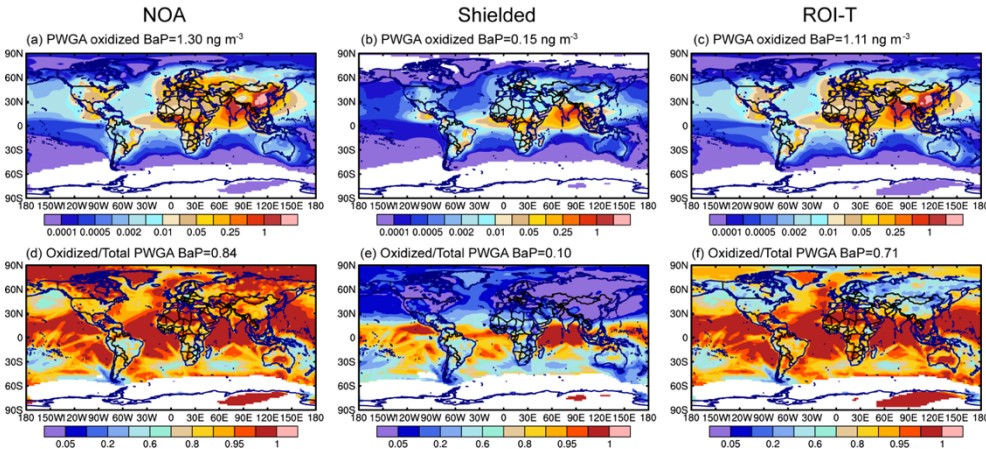

**Figure 3. The spatial distribution of simulated (a-c) oxidized BaP concentrations and (d-f) the ratio of oxidized BaP to the total (fresh+oxidized) BaP in DJF. Simulations with the different heterogeneous reactions of particle-bound BaP approaches are shown in the left (NOA simulation), middle (Shieled), and right (ROI-T) columns, respectively.**

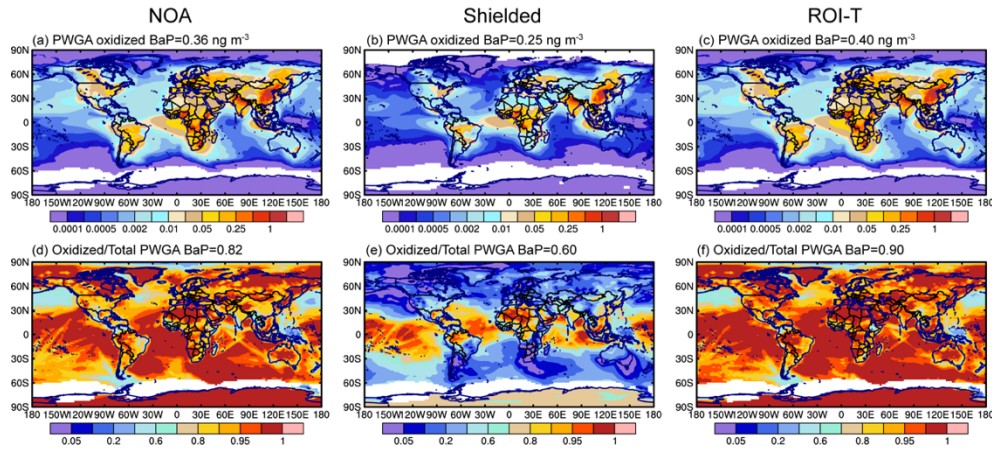

**Figure 4. Same as Figure 3 but for JJA.**

Due to the less effectiveness of OA coatings under warm and moist conditions, all simulations with
different BaP degradation approaches predict that oxidized BaP contributes to more than 90% of the total
BaP concentrations in JJA in the tropics (30°S-30°N). At mid-to-high latitudes, the oxidized BaP varies
greatly with the effect of OA coatings on the BaP oxidation reactions. More than 80% of total BaP is
oxidized in mid-to-high latitudes applying the NOA or ROI-T approaches, with peaks exceeding WHO
recommendations in most of East Asia, West Europe, and North America. Meanwhile, the Shielded
approach assumes that OA coatings largely limit the particle-bound BaP oxidation reaction, whereas the
oxidized BaP contributes no more than 40% to the total BaP. Our results indicate that current model
estimates of human exposure to fresh or oxidized PAHs are highly sensitive to assumptions about PAH
degradation processing, especially during North Hemisphere winter.
**3.3. Model Evaluation**
**3.3.1 Fresh BaP**
To assess simulated BaP concentrations, we select surface BaP measurements from 66
background/remote sites and 208 urban-affected sites worldwide (Table S2, S3), covering the period
1997 to 2014, with a focus on the years between 2004 and 2011. Median BaP observations at each site
are compared with simulated BaP in DJF and JJA, respectively. Given the global model's horizontal grid
spacing of approximately 200 km, we specifically compare simulated BaP concentrations with
measurements from background sites. To address the limitations of the coarse global model, we
downscale the simulated BaP grid to a higher resolution of 10 km based on factors such as wind speed,
wind direction and frequency, emission density, and gas/particle BaP degradation rates to account for
strong gradients and high BaP concentrations near urban areas. This downscaled approach aims to
account for strong gradients and high BaP concentrations near source areas. Our previous studies, using
the same simulations, reported that while the coarse-grid model significantly underestimates
concentrations in urban-affected regions, the downscaled BaP vastly improves the comparison between
the model and observations [Shrivastava et al., 2017; Lou et al., 2023].
Figure 5a compares measured and model-predicted concentrations at 66 background sites around the
world in DJF. The model-estimated BaP for the Shielded approach during the same time and locations
of the measurements agrees best with observations of global BaP concentrations, with a normalized mean
bias (NMB) of -18%. In contrast, without the effect of OA coating on the degradation of particle-bound
BaP, NOA predictions are 78% lower than observed BaP globally (Fig. 5b). Comparisons between
measured and downscaled simulated BaP at urban-affected sites show similar results, as the OA shielding
approach significantly improves the model's ability to predict fresh BaP concentrations.
It's worth noting that the effectiveness of OA coatings depends largely on temperature and humidity,
which are related to the meteorological characteristics of different regions. We, therefore, compare
measured and model-simulated BaP concentrations at different latitudes, namely relatively high latitudes
(measured locations north of 40°N) and low latitudes (measured locations between 40°S and 40°N),
respectively. Figures 6a and 6b demonstrate that the OA shielding particle-bound BaP approach increases
the simulated BaP concentrations in much better agreement with the measured values than without the
OA coating effect. This improvement is not sensitive to latitude. For the ROI-T treatment, although
predicted fresh BaP concentrations at locations above 40°N were two or three times higher than the
treatment without OA coating effects, the simulation still substantially underestimates the BaP
concentrations in these regions by 50% (Fig. 6c). Moreover, model-estimated BaP concentrations in
ROI-T perform even worse at low latitudes compared to high latitudes. On a global average, the ROI-T
approach, accounting for the temperature and humidity dependence of the phase state, diffusivity, and
reactivity of particulate-bound BaP, underestimates BaP by ~60% in DJF (Fig. 5a, c).

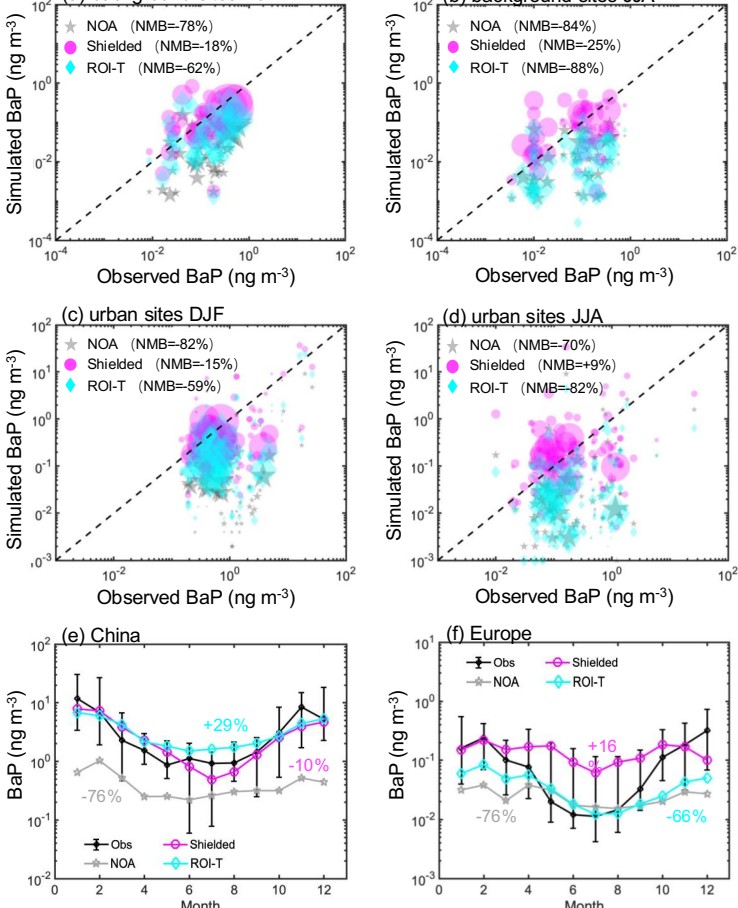


**459    Figure 5. Comparison between simulated surface BaP concentrations from NOA, Shielded, and ROI-T**

**460    simulations and ground-based measurements (a-b) for 66 background sites, (c-d) for 208 urban impacted sites.**

**461    The circle area is proportional to the number of days sampled at each site. Annual variation of measured and**

**462    simulated BaP concentrations at (e) 18 sites (6 background and 12 urban impacted sites) in China, and (f) 18**

**463    background sites in Europe. Black lines represent measured values (median and 15th and 85th percentiles of**

**464    site monthly means), while blue (NOA), red (Shielded), and green (ROI-T) lines represent the median of the**

**465    monthly model-simulated means for these sites.**


In JJA, both NOA and ROI-T simulations struggle to capture BaP concentrations, underestimating
observations by more than 70% (Fig. 5b and 5d). However, similar to DJF, the model-predicted
concentrations in simulation using the Shielded approach exhibit the best agreement with ground
measurements in JJA, showing a NMB of -25% and +9% at the background and urban-affected sites,
respectively. Interestingly, the ROI-T approach deviates more from the actual observed values, especially
at measurement locations between 40°N and 40°S (Fig. 6d and 6f).
Figure 5e indicates that both the Shielded simulation (red line) and ROI-T simulation (green line) capture
the magnitude and seasonal variations of BaP concentrations compared with monthly observations at 18
sites in China (black lines). The regions of China and Europe were chosen for this analysis because they
meet two criteria: (1) each has more than 10 measurement sites, and (2) the data from these sites span
over one year, allowing for a reliable representation of seasonal variations. In both simulations and
observations, BaP concentrations peak in winter but are lowest in summer. As mentioned in section 3.1,
the predicted monthly variations in BaP concentrations are due to the seasonality of BaP emissions and
BaP oxidation rates. For instance, residential emissions in China are four times higher in winter than in
summer, contributing 78% of BaP emissions in winter and 56% in summer [Shen et al., 2013].
Furthermore, lower wintertime temperatures favor more viscous OA coatings to reduce BaP diffusion
and decrease oxidation rates, while more liquid-like OA coatings in summer have a minor effect on BaP
oxidation reactions. In contrast, although the models show a similar seasonal cycle to observations, fresh
BaP concentrations are largely underestimated throughout the year in the absence of the OA coating
status effect (NOA).

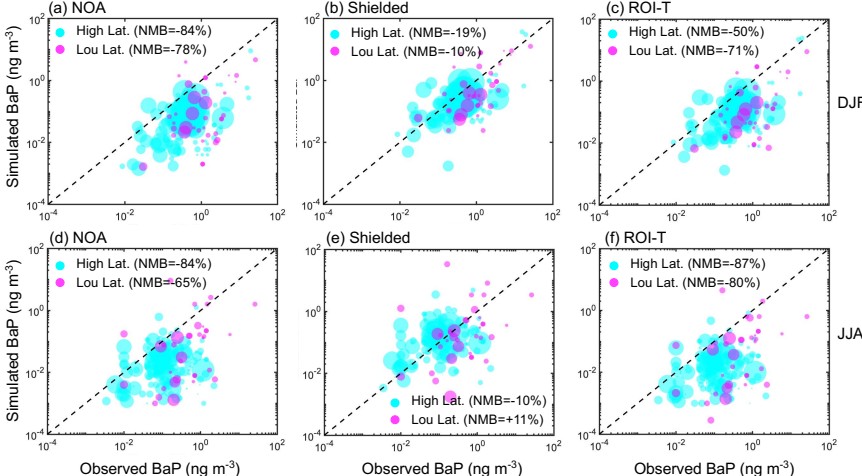


**Figure 6. Comparison between simulated surface BaP concentrations at relatively high latitude regions**
**(marked as olive, measured locations north of 40°N) and low latitude regions (marked as magenta, measured**
**locations between 40°N and 40°S) in (a-c) DJF and (d-f) JJA, respectively. The circle area is proportional to**
**the number of days sampled at each site. Both background and urban sites are included.**

In Europe, although the simulated BaP for the Shielded approach also exhibits the best agreement with
observations at 18 sites throughout the year, with a NMB of +16%, the simulated BaP fails to capture
the magnitude of the measured BaP concentrations during the warm season (Fig. 5f). From April to
October, model-predicted BaP concentrations in the Shielded simulation are overestimated by 88%. In
contrast, the simulated BaP concentrations for the ROI-T approach are consistent with the monthly
variation of the measured BaP concentrations, despite showing a 66% underestimated annual mean,
which is mainly due to the significant underestimation in cool season. Our results suggest that while the
Shielded simulation is likely closer to actual BaP magnitudes at mid- and low-latitudes, the ROI-T
approach may better represent seasonal variation at mid-and high-latitudes but overestimates the
coefficient of BaP multiphase degradation rates.
**3.3.2 Oxidized BaP**
Due to limited observations of oxidized BaP, specifically 1-, 3-, and 6-nitrobenzo(a)pyrene, we assess
monthly changes in BaP for three different particle-bound BaP degradation approaches performed at two
sites, Grenoble and SIRTA. In this study, we compare both simulated fresh BaP and oxidized BaP with
in situ measurements. For Grenoble, we use downscaled simulated fresh/oxidized BaP concentrations
for comparison, while at the SIRTA site, we compare coarse-resolution simulated BaP concentrations
with measurements.

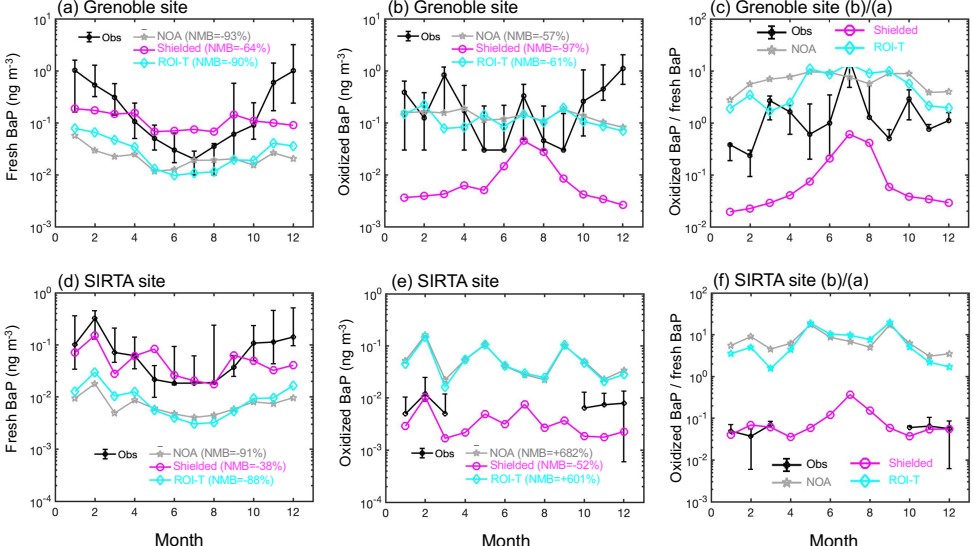

**Figure 7. Monthly comparison between simulated surface fresh and oxidized BaP from NOA, Shielded, and**
**ROI-T simulations and ground-based measurements at (a-c) Grenoble and (d-f) SIRTA sites. Black lines**
**represent measured values (median and 15th and 85th percentiles of each site), while blue (NOA), red**
**(Shielded), and green (ROI-T) lines represent the model-simulated median per month. The ratio of oxidized**
**BaP to fresh BaP are represent in (c) and (f). From April to September at SIRTA site, observed concentrations**
**of oxidized BaP were below LQ and therefore not presented on the graph.**

The measurement site at the sampling station of "Les Frenes" in Grenoble (5.73ºE, 45.16ºN, France)
represents the most densely populated urban area in Europe. Although the simulated concentrations
applying the Shielded approach best match the observed fresh BaP concentrations in Grenoble among
the three approaches, the model largely underestimates winter BaP concentrations but overestimates
summer concentrations (Fig. 7a). Therefore, the assumption that viscous organic aerosol coatings
completely shut off the reaction of fresh particle-bound BaP with ozone under cool and dry conditions
is somewhat distorted and fails to capture seasonal variation in fresh BaP in Grenoble. The relatively low
ratio of oxidized to fresh BaP using the Shielded approach in Fig. 7c indicates that the chemical scheme
overly protects fresh PAHs from oxidation. Consequently, it underestimates oxidized BaP by one order
of magnitude. In contrast, despite the overall underestimation, the ROI-T simulation captures the
seasonal variations in fresh BaP concentrations in Grenoble (Fig. 7a). However, the magnitude of the
simulated oxidized BaP concentration is very similar to the observed values (Fig. 7b). As we mentioned
above, the ratio of oxidized to fresh BaP in Fig. 7c reveals that the oxidation rate of BaP from fresh to
oxidized is too fast under ROI-T treatment, especially during cold season.
In addition, the underestimation of both fresh and oxidized BaP concentrations may be partly due to the
coarse horizontal resolution of simulated BaP, and inaccurate urban PAH emissions. We use a
downscaling formulation to convert the 200 km grid resolution to a ~10 km grid resolution, but the spatial
distribution of BaP obtained in this way is highly dependent on accurate emissions and meteorological
fields. Previous studies have reported that traditional biomass combustion for residential heating is the
main source of $PM_{2.5}$ in France in winter and including in the Grenoble area [Favez et al., 2009; 2021;
Srivastava et al., 2018; Weber et al., 2019; Zhang et al., 2020a], thus inevitably emitting large amounts
of BaP. Considering the underestimation of both fresh and oxidized BaP concentrations at the Grenoble
site in winter, there is a large uncertainty in the emission and spatial distribution of PAHs in urban areas
(Fig. 7a, b).
For the SIRTA site, the simulated BaP from the Shielded simulation shows good performance compared
to the observed concentrations of fresh and oxidized BaP in winter (Fig. 7d, e). The performance of
Shielded approach in summer remains unclear due to the lack of observed concentrations of oxidized
BaP (Fig. 7e). However, the underestimation of fresh BaP concentrations and overestimation of oxidized
BaP concentrations in ROI-T and NOA suggest that the particle-bound PAH degradation rate is too fast
for these two approaches (Fig. 7c-d).
According to the ROI-T approach, once BaP is absorbed by organic aerosols, it can only be oxidized
when it comes to the surface through bulk diffusion or $O_3$ absorption from the gas sorption layer to bulk
layers. The changes in the BaP degradation rate coefficient are highly dependent on variations in
temperature and relative humidity [Mu et al., 2018]. Considering that RH in the French winter is
generally higher than 70% (Fig. 2b), the BaP degradation rate coefficient decreases by only one order of
magnitude for every 20 K drop in temperature from around 293 K (Table S1). Therefore, the oxidation
rate of ROI-T for particle-bound PAHs is reduced by no more than 50% when the temperature is around
280 K in the French winter (Fig. 2a). Our results suggest that at higher humidity, the ROI-T approach
underestimates the impact of OA coatings on PAH degradation effectiveness. Thus, the model's ability
to simulate fresh BaP is not significantly enhanced over the default NOA when the ROI-T approach is
selected, as relative humidity is significantly higher than 70% in mid- and high-latitude winters (Figs. 5f,
6c).
**3.4. Lung-cancer risk of PAH mixture**
As an indicator of cancer risk from PAH mixtures, previous studies calculated PAH-associated health
risks based on exposure to BaP concentrations using a method grounded in epidemiological data
[Bostrom et al., 2002; Zhang et al., 2009; Shen et al., 2014; Shrivastava et al., 2017; T Wang et al., 2017;
Kelly et al., 2021]. These studies primarily considered fresh PAHs when assessing PAH-associated
health risks. In this study, we follow the approach of previous studies to estimate ILCR [Shen et al., 2014;
Shrivastava et al., 2017; Lou et al., 2023].
Figure 8a illustrates that global and regional population-weighted ILCR varies significantly across
simulations when only considering exposure to fresh PAH. This variation is due to the substantial impact
of PAH degradation approaches on fresh BaP concentrations. On a global population-weighted basis, the

ILCR is predicted to be ~$0.6 \times 10^{-5}$ from the NOA and ROI-T simulations, falling within WHO-acceptable risk levels for PAH exposure. However, based on the Shielded simulation, the global population-weighted ILCR is predicted to be ~$2 \times 10^{-5}$, exceeding the acceptable limit of 1 death per 100,000 persons. Moreover, without the heterogeneous oxidation of BaP, Shen et al. (2014) predicted an even higher global population-weighted ILCR of $3 \times 10^{-5}$. These results underscore the high sensitivity of global ILCR estimates to the choice of PAH degradation approaches.

The variations in fresh BaP exposure and population-weighted ILCR are even more important for regional estimation. Using the Shielded approach, the regional average population-weighted ILCR is predicted to exceed $1 \times 10^{-5}$ over East Asia, South Asia, Southeast Asia, Russia, Africa, and South America. In contrast, ILCR for NOA and ROI-T simulations suggests a 3-4 times lower lung cancer risk in these regions, expected to be below $1 \times 10^{-5}$ except in East Asia (Fig. 8a). Due to the high emission levels in 2008, the ROI-T simulation estimates a 50% higher ILCR than the NOA simulation, also exceeding the WHO acceptable limit in the East Asia.

Furthermore, recent laboratory studies suggest that oxidized PAHs persist in the particle-phase and often appear as higher molecular weight peaks in particle mass spectra [Ringuet et al., 2012; Zelenyuk et al., 2012; Keyte et al., 2013; Jariyasopit et al., 2014]. Some PAH oxidation products may even be more toxic than their parent compounds [EHC 2003; Clergé et al., 2019; Hrdina et al., 2022; Peng et al., 2023]. A quantitative understanding of the toxicity of these products is lacking, as each parent PAH could be oxidized into hundreds of products. In this study, we conduct a conservative first-order calculation of lung cancer risk associated with oxidized PAHs, assuming that PAH oxidation products have the same toxicity as their parents. On a global population-weighted basis, the ILCR is projected to 2.5 deaths per 100,000 persons when exposure to oxidized BaP is added to our previous calculations of fresh BaP exposure for all three simulation approaches (Fig. 8b). While three-quarters of the global population-weighted ILCR for the Shielded simulation is contributed to fresh PAH, oxidized PAHs contribute approximately 40% the ILCR in warm and humid regions such as South Asia, Southeast Asia, and Africa. In comparison, NOA and ROI-T simulations predict a dominant contribution of ILCR from PAH oxidation products compared to fresh PAHs over most regions of the globe. For example, over East Asia and South Asia, the NOA and ROI-T simulations predict that the regional population-weighted ILCR will exceed 3 deaths per 100,000 persons resulting from the oxidized PAHs alone, compared to ~1 death per 100,000 persons from oxidized PAHs in the Shielded simulation. The oxidized-fresh PAH ILCR split is much greater in the NOA and ROI-T simulations compared to the Shielded simulation.

Despite differences in organic coating effectiveness and heterogeneous reactivity between the NOA, Shielded, and ROI-T simulations, all schemes suggest that oxidized PAHs are crucial for lung cancer risk and cannot be neglected. If the toxicity of oxidized PAHs is similar to fresh PAHs, the total ILCR (fresh+oxidized) is comparable in the three approaches. However, oxidized PAHs could be much more important in certain regions (such as Southeast Asia, South Asia, and Africa), depending on their composition/toxicity and where the organic coatings are less effective in shielding them from heterogeneous reactivity. Considering the high levels of oxidized PAHs in mid-to-low latitudes, the measurements for oxidized PAHs, as well as human health exposure to oxidized PAHs, are necessary for further studies [Kelly et al., 2021].

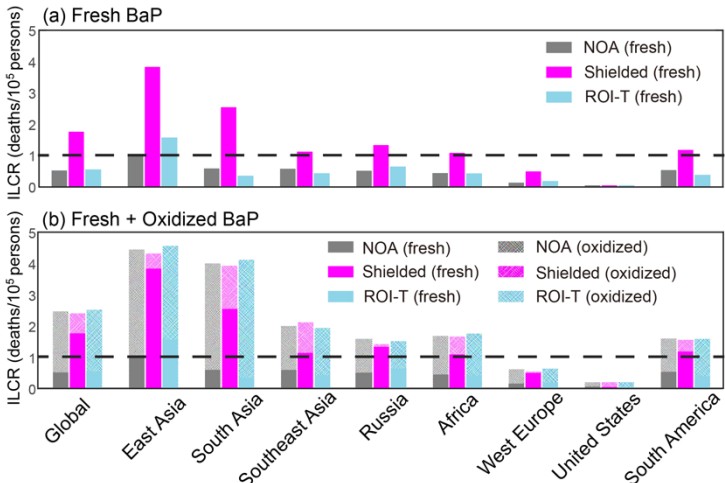

613

**Figure 8. Regional PAH-associated ILCR from NOA, Shielded, and ROI-T simulations. Note that solid bars in (a, b) and shaded bars in (b) represent ILCR calculated from exposure to fresh and oxidized PAHs, respectively.**

## 4. Conclusion and Discussion

This study uses the CAM5 model to investigate the impact of particle-bound PAH degradation approaches on the spatial distribution of BaP, considering the presence or absence (or lack of effect) of OA coatings. The three PAH degradation approaches are (1) no effect of OA coating state (e.g., thickness) on BaP oxidation (NOA), (2) thick OA coatings (> 20 nm thick) completely shielding particle-bound BaP oxidation under cool, dry conditions (Shielded), and (3) OA coatings slowing down particle-bound BaP oxidization in response to temperature and humidity (ROI-T). The results show significant seasonal variations in BaP concentrations driven by emissions, deposition, and the chosen degradation approach. In DJF, without the viscous OA coatings, the model severely underestimates the global fresh BaP concentration (PWGA=0.24 ng m$^{-3}$), as 82% of the BaP is oxidized near source areas. Including viscous OA coatings substantially slows oxidation, leading to 2-6 times higher fresh BaP concentrations. Notably, the Shielded approach predicts the highest BaP concentration. While the Shielded and ROI-T approaches predict similar fresh BaP distributions at high latitudes, the ROI-T approach suggests faster heterogeneous degradation kinetics of BaP with ozone at warm temperatures (>288K, see Table S1), resulting in lower predicted BaP concentrations in regions such as Africa, South Asia, and East Asia. In JJA, BaP concentrations are concentrated near source areas. The Shielded approach continues to predict higher fresh BaP concentrations than simulations without viscous OA coatings, while the ROI-T approach predicts the lowest concentrations due to faster oxidation under warmer, more humid conditions. Over 90% of BaP is oxidized in the tropics (30°S-30°N) in all simulations. At mid-to-high latitudes, oxidation rates vary greatly depending on OA coating assumptions. In the Shielded simulation, only 40% of BaP is oxidized, compared to 91% in the NOA and 94% in the ROI-T simulations. Model performance is compared with observations at 66 background/remote and 208 non-background sites globally. The NOA and ROI-T simulations perform poorly, underestimating fresh BaP concentrations by 60-80%. However, the Shielded approach agrees best with observations, with a NMB consistently within ±20%. Both the Shielded and ROI-T approaches improve BaP concentrations and

seasonal variations in China. While the Shielded approach aligns more closely with BaP concentrations in Europe, it fails to capture seasonal variation in fresh BaP. Additionally, oxidized BaP concentrations from Grenoble (2013) and SIRTA (2014-2015) are also used for model evaluation. Our results indicate that the Shielded approach underestimates BaP oxidation rates, particularly in the warm season, while the ROI-T approach overpredicts oxidation rates in multiple regions but better represents seasonal variations in fresh BaP concentrations.

Epidemiological data are used to estimate the population-weighted ILCR associated with PAH exposure. For fresh BaP alone, the PWGA ILCR ranges from $\sim 0.6 \times 10^{-5}$ to $\sim 2 \times 10^{-5}$ among the three simulations, exceeding the acceptable limit of 1 death per 100,000 persons when considering the effect of viscous OA coatings. When both fresh and oxidized PAHs are assumed to equally contribute to cancer risk, the total ILCR remains comparable across all simulations, amounting to 2.5 deaths per 100,000 persons. This is because the combined concentration of fresh and oxidized BaP is similar across the three simulations. This highlights the potential significance of oxidized PAHs, particularly in regions where they may play a substantial role in health risks.

This study highlights the importance of understanding the effects of viscous OA coatings on BaP degradation. However, the approaches used in this study have limitations that need to be addressed in future research, to better improve global PAH simulations. The viscosity of OA affects BaP degradation in two ways: (1) through gas-particle partitioning of SOA, and (2) by altering BaP degradation kinetics with ozone. In terms of gas-particle partitioning, viscosity is treated differently in each approach. In the NOA simulation (section 2.3), the FragSVSOA treatment is used, which assumes SOA remains a semi-volatile, liquid-like solution throughout its lifetime, without considering the effect of its viscosity on gas-particle partitioning. In contrast, the Shielded and ROI-T approaches use the FragNVSOA treatment, which assumes that SOA rapidly transitions to a highly viscous, non-volatile semi-solid within 30 minutes due to paricle-phase oligomerization reactions [Shrivastava et al., 2015]. Any further gas-phase organic oxidation products do not form a solution with pre-existing OA. This assumption aligns with a recent experimental study showing that isoprene SOA undergoes rapid aging (~20 minutes), leading to oligomer and organosulfate formation and a phase transition to semi-solid SOA [Chen et al. 2023].

Regarding the second effect, both the Shielded and ROI-T approaches incorporate the effect of SOA viscosity on BaP degradation kinetics with ozone. The Shielded approach, based on Shrivastava et al. (2017), assumes that thick OA coatings completely shield particle-bound BaP oxidation under cool (temperature < 296 K) and dry (RH < 50%) conditions, thus providing an upper bound of the impact of viscous SOA on BaP degradation and underestimating the oxidation rate, partially in urban areas. The ROI-T approach models the multiphase degradation of BaP with ozone, incorporates both mass transport and chemical reactions of particle-bound species in the bulk phase and at the surface. The first-order decay rates of BaP, as presented in Table S3 of Mu et al. (2018), are parameterized values that already account for the impact of changing SOA viscosity as a function of temperature and RH on BaP degradation, although this may overestimate the oxidation rate in remote area (Fig. 7).

Both the Shielded and ROI-T approaches assume a globally constant 30-minute timescale for the transformation of SOA to a semi-solid state, which aligns with a recent experimental study [Chen et al. 2023]. However, future studies are needed to measure the phase transition timescale of different SOA types under varying temperature and relative humidity conditions. Previous modelling studies have suggested that the phase state of OA varies significantly with environmental factors such as temperature,

RH in different SOA systems [Pye et al., 2017; Shiraiwa et al., 2017; Zhang et al., 2019; Li et al., 2020; Schmedding et al., 2020]. In addition, water associated with organics has been suggested to be the primary predictor of OA viscosity [Rasool et al., 2021]. The effects of phase separation within SOA-water mixtures and variability in the water uptake ability of SOA as a function of its aging and gas-particle partitioning, and the resulting impacts on SOA viscosity and BaP reaction kinetics need to be considered in future studies.

The pp-LFER approach used in our study considers the effects of BaP partitioning to a two-phase organic system consisting of liquid water-soluble/organic soluble organics, and solid/semisolid organic polymers. In this sense, this approach considers the impacts of water-soluble organics and organic polymers (formed by accretion reactions) on BaP partitioning. Further experiments that measure the partitioning of BaP on liquid-like SOA and polymeric SOA systems are needed to constrain pp-LFER model predictions.

To improve global PAH simulations, future research should focus on understanding the impact of OA coatings on PAH degradation effectiveness, particularly incorporating the variations in phase-state of OA coatings in models. Expanding the observational dataset to include a wider range of ground-based and satellite-derived measurements, such as water-soluble organic aerosols (Zhang et al., 2020b), will also be crucial for validating and refining these models. Additionally, further exploration of the chemical composition and toxicity of both fresh and oxidized PAHs is necessary to assess their role in air quality and human health risk more effectively.

**Data availability**

The ERA-Interim reanalysis data is available from https://www.ecmwf.int/en/forecasts/datasets/reanalysis-datasets/era-interim. The GEMS 0.1º × 0.1º global BaP emission inventory is available from gems.sustech.edu.cn. The long-term observation data are obtained from IADN (https://www.epa.gov/great-lakes-monitoring/great-lakes-integrated-atmospheric-deposition-network), EMEP (https://www.emep.int), and GENASIS (https://www.genasis.cz).

**Acknowledgments**

This research was supported by the National Natural Science Foundation of China (grant number: 42075095), Fundamental Research Funds for the Central Universities (grant number: 14380198), the Energy Exascale Earth System Model (E3SM) project. Manish Shrivastava was supported by the U.S. Department of Energy (DOE) Office of Science, Office of Biological and Environmental Research (BER) through the Early Career Research programme and DOE BER's Atmospheric System Research (ASR) program. The Pacific Northwest National Laboratory (PNNL) is operated for DOE by Battelle Memorial Institute under Contract DE-AC05-76RL01830. This work was also supported by the French Ministry of environment and the National reference laboratory for air quality monitoring in France (LCSQA).

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
