# Peer review of "Uncertainties in the effects of organic aerosol coatings on"

_EGUsphere, 2024_

## Referee Comment (RC2)

**Review for "Uncertainties in the effects of organic aerosol coatings on polycyclic aromatic hydrocarbon concentrations and their estimated health effects" by Lou et al.**

In this study the authors compared three BaP degradation approaches with different treatment of the effects of organic aerosol coatings on the particle-phase BaP reactivity. The simulated gas-phase BaP and oxidized BaP based on the three approaches were compared with observations. As BaP has significant health effects, accurate representation of BaP concentrations over a global scale is highly needed. This manuscript is clearly written and is helpful for future modelling studies to improve the PAHs simulations. I recommend the publication of this work after the authors could address my comments below.

**Major comments:**

(1) My major concern is that how to simulate the effects of organic aerosol coating, namely, the viscosity, on the gas-particle partitioning of BaP and the heterogeneous degradation coherently. Viscosity affects both partitioning and reactivity. Therefore, if the Bap degradation in the particle phase has considered the effects of viscosity in the Shielded approach and ROI-T approach as what the author did, for coherence, the effect of viscosity on gas-particle partitioning of BaP should be also included. Does the ppLFER model the authors applied for gas-particle partitioning of BaP specifically considered the viscosity of the organic shell and its effect on partitioning?

(2) My another concern is that the section of "Conclusion and Discussion" is too much rephrasing the results with little discussion. I recommend that the authors can give a clearer conclusion. For the discussion, the following aspects may be helpful. First, what are the limitations of the Shielded and ROI-T approaches and how to improve these kinds of methods for future modelling studies? For example, Bap and ROI-T have not considered the phase state of the organic coatings varied with RH, $T$, and organic aerosol chemical composition, which, however, has been parameterized since 2017 (Li et al., 2020; Shiraiwa et al., 2017; Zhang et al., 2019) and included in chemical transport models (Shrivastava et al., 2022; Zhang et al., 2024). Second, would phase phase separation affect BaP gas-particle partitioning and heterogeneous degradation (Pye et al., 2017; Schmedding et al., 2020)?

**Specific comments:**

(1) Some recent modeling work on PAH should be cited and the simulated BaP concentrations are better compared (Wu et al., 2024).

(2) Line 164 at Page 5, why the authors did not include the gas-phase reaction of BaP with $O_3$ and $NO_3$?

(3) Line 348-350 at Page 11, the authors wrote that a large portion of the total Bap was still oxidized though the effect of organic coating had been considered in bulk diffusion of BaP in the ROI-T scheme, which was surprising. Again, referring to my major comment 1, the authors should clarify that the effect of viscosity on

gas-particle partitioning and BaP degradation rate is related to two different processes. Did the ROI-T scheme consider the phase state effect on bulk diffusivity of BaP in your simulation? I think the result showed in Line 348-350 is not surprising as the reaction rate of BaP with ozone in the particle phase was dependent on RH and T, thus the viscosity of organic shell, which resulted in limited effect of organic coating on BaP degradation in areas with low viscous particles.

(4) Line 198-200 on Page 6, please double check. I did not see this from Table S3 in Mu et al. (2018).

**References**
1. Li, Y., Day, D. A., Stark, H., Jimenez, J. L., and Shiraiwa, M.: Predictions of the glass transition temperature and viscosity of organic aerosols from volatility distributions, Atmos. Chem. Phys., 20, 8103-8122, 10.5194/acp-20-8103-2020, 2020.
2. Pye, H. O. T., Murphy, B. N., Xu, L., Ng, N. L., Carlton, A. G., Guo, H., Weber, R., Vasilakos, P., Appel, K. W., Budisulistiorini, S. H., Surratt, J. D., Nenes, A., Hu, W., Jimenez, J. L., Isaacman-VanWertz, G., Misztal, P. K., and Goldstein, A. H.: On the implications of aerosol liquid water and phase separation for organic aerosol mass, Atmos. Chem. Phys., 17, 343-369, 10.5194/acp-17-343-2017, 2017.
3. Schmedding, R., Rasool, Q. Z., Zhang, Y., Pye, H. O. T., Zhang, H., Chen, Y., Surratt, J. D., Lopez-Hilfiker, F. D., Thornton, J. A., Goldstein, A. H., and Vizuete, W.: Predicting secondary organic aerosol phase state and viscosity and its effect on multiphase chemistry in a regional-scale air quality model, Atmos. Chem. Phys., 20, 8201-8225, 10.5194/acp-20-8201-2020, 2020.
4. Shiraiwa, M., Li, Y., Tsimpidi, A. P., Karydis, V. A., Berkemeier, T., Pandis, S. N., Lelieveld, J., Koop, T., and Pöschl, U.: Global distribution of particle phase state in atmospheric secondary organic aerosols, Nat. Commun., 8, 15002, 10.1038/ncomms15002, 2017.
5. Shrivastava, M., Rasool, Q. Z., Zhao, B., Octaviani, M., Zaveri, R. A., Zelenyuk, A., Gaudet, B., Liu, Y., Shilling, J. E., Schneider, J., Schulz, C., Zöger, M., Martin, S. T., Ye, J., Guenther, A., Souza, R. F., Wendisch, M., and Pöschl, U.: Tight Coupling of Surface and In-Plant Biochemistry and Convection Governs Key Fine Particulate Components over the Amazon Rainforest, ACS Earth and Space Chemistry, 6, 380-390, 10.1021/acsearthspacechem.1c00356, 2022.
6. Wu, Z., Chen, X., Wang, Z., Chen, H., Wang, Z., Mu, Q., Wu, L., Wang, W., Tang, X., Li, J., Li, Y., Wu, Q., Wang, Y., Zou, Z., and Jiang, Z.: Modeling of polycyclic aromatic hydrocarbons (PAHs) from global to regional scales: model development (IAP-AACM_PAH v1.0) and investigation of health risks in 2013 and 2018 in China, Geosci. Model Dev., 17, 8885-8907, 10.5194/gmd-17-8885-2024, 2024.
7. Zhang, Y., Nichman, L., Spencer, P., Jung, J. I., Lee, A., Heffernan, B. K., Gold, A., Zhang, Z., Chen, Y., Canagaratna, M. R., Jayne, J. T., Worsnop, D. R., Onasch, T. B., Surratt, J. D., Chandler, D., Davidovits, P., and Kolb, C. E.: The Cooling Rate and Volatility Dependent Glass Forming Properties of Organic Aerosols Measured by Broadband Dielectric Spectroscopy, Environmental Science & Technology, 10.1021/acs.est.9b03317, 2019.
8. Zhang, Z., Li, Y., Ran, H., An, J., Qu, Y., Zhou, W., Xu, W., Hu, W., Xie, H., Wang, Z.,

Sun, Y., and Shiraiwa, M.: Simulated phase state and viscosity of secondary organic aerosols over China, Atmos. Chem. Phys., 24, 4809-4826, 10.5194/acp-24-4809-2024, 2024.

---

## Author Comment (AC1)

Review 2 for "Uncertainties in the effects of organic aerosol coatings on polycyclic aromatic hydrocarbon concentrations and their estimated health effects" by Lou et al.

In this study the authors compared three BaP degradation approaches with different treatment of the effects of organic aerosol coatings on the particle-phase BaP reactivity. The simulated gas-phase BaP and oxidized BaP based on the three approaches were compared with observations. As BaP has significant health effects, accurate representation of BaP concentrations over a global scale is highly needed. This manuscript is clearly written and is helpful for future modelling studies to improve the PAHs simulations. I recommend the publication of this work after the authors could address my comments below.

Thank you for your valuable comments and suggestions. We have carefully addressed each point as follows.

Major comments:
(1) My major concern is that how to simulate the effects of organic aerosol coating, namely, the viscosity, on the gas-particle partitioning of BaP and the heterogeneous degradation coherently. Viscosity affects both partitioning and reactivity. Therefore, if the Bap degradation in the particle phase has considered the effects of viscosity in the Shielded approach and ROI-T approach as what the author did, for coherence, the effect of viscosity on gas-particle partitioning of BaP should be also included. Does the ppLFER model the authors applied for gas-particle partitioning of BaP specifically considered the viscosity of the organic shell and its effect on partitioning?

Response:

Thank you for your insightful comment. As you correctly pointed out, viscosity of the organic aerosol coating is an important factor that could influence both BaP gas-particle partitioning and the heterogeneous degradation reactivity.

In this study, the pp-LFER approach considers the effects of BaP partitioning to a two-phase organic system consisting of liquid water-soluble/organic soluble organics, and solid/semisolid organic polymers. In this sense, this approach considers the impacts of water-soluble organics and organic polymers (formed by accretion reactions) on BaP partitioning. However, further experiments that measure the partitioning of BaP on liquid-like SOA and polymeric SOA systems are needed to constrain pp-LFER model predictions.

The viscosity of OA affects the atmospheric lifecycle of BaP in two ways: (1) through gas-particle partitioning of SOA that also impacts gas-particle partitioning of BaP (pp-

LFER approach), and (2) by altering BaP heterogeneous degradation kinetics with ozone. From the first perspective, we handle viscosity differently across the three approaches. In the NOA simulation, we use the FragSVSOA treatment described in Shrivastava et al. (2015), assuming SOA as semi-volatile liquid-like well-mixed solution throughout its atmospheric lifetime, which does not consider the effects of viscosity on gas-particle partitioning of SOA. In contrast, the Shielded and ROI-T approaches use the FragNVSOA treatment, which assumes that SOA, once formed, is transformed to a highly viscous non-volatile semi-solid SOA within the same global model timestep (30 min) due to particle-phase oligomerization reactions within SOA [Shrivastava et al., 2015]. The FragNVSOA treatment also assume that any further gas-phase organic oxidation products that condense do not form a solution with pre-existing OA. This assumption is consistent with recent experimental studies, which show a short aging timescale of ~20 min that causes oligomer and organosulfate formation within isoprene SOA causing non-equilibrium partitioning behavior and phase transition to semi-solid SOA [Chen et al., 2023].

From the second perspective, both the Shielded and ROI-T approaches incorporate the effects of SOA viscosity on BaP degradation kinetics with ozone. The Shielded approach, based on Shrivastava et al. (2017), assumes that thick OA coatings (> 20 nm) completely shield particle-bound BaP oxidation under cool (temperature < 296 K) and dry (RH < 50%) conditions, thus providing an upper bound of the impact of viscous SOA on BaP degradation, however this approach moderately underestimates the oxidation rate in urban areas. The ROI-T approach models the multiphase degradation of BaP with ozone, incorporates both mass transport and chemical reactions of particle-bound species in the bulk phase and at the surface. The first-order decay rates of BaP, as presented in Table S3 of Mu et al. (2018), are parameterized values that already account for the impact of changing SOA viscosity as a function of temperature and RH on BaP degradation, though this may overestimate the oxidation rate in remote areas (Fig. 7).

Our  assumption of a globally constant 30-minute particle-phase reaction and oligomerization timescale for SOA (converting it to semi-solid SOA) is consistent with the recent experimental study [Chen et al., 2023]. However, future studies are needed to measure the phase transition timescale of different SOA types under varying temperature and relative humidity conditions. Previous modeling studies have suggested that the phase state of OA varies significantly with environmental factors such as temperature, RH in different SOA systems [Pye et al., 2017; Shiraiwa et al., 2017; Zhang et al., 2019; Li et al., 2020; Schmedding et al., 2020]. However, experimental measurements of the phase transition timescales of atmospherically relevant SOA are needed, similar to the study by Rasool et al. (2021). In addition, water associated with organics has been suggested to be the primary predictor of OA viscosity [Rasool et al., 2021] .The effects of phase separation within SOA-water mixtures and variability in the water uptake ability of SOA as a function of its aging and  gas-particle partitioning, and the resulting impacts on BaP reaction kinetics need to be considered in future studies.

We acknowledge that incorporating the viscosity of the organic shell could provide a more comprehensive representation of gas-particle partitioning, and we have revised the "conclusion and discussion" section to address this comment.

(2) My another concern is that the section of "Conclusion and Discussion" is too much rephrasing the results with little discussion. I recommend that the authors can give a clearer conclusion. For the discussion, the following aspects may be helpful. First, what are the limitations of the Shielded and ROI-T approaches and how to improve these kinds of methods for future modelling studies? For example, Bap and ROI-T have not considered the phase state of the organic coatings varied with RH, T, and organic aerosol chemical composition, which, however, has been parameterized since 2017 (Li et al., 2020; Shiraiwa et al., 2017; Zhang et al., 2019) and included in chemical transport models (Shrivastava et al., 2022; Zhang et al., 2024). Second, would phase phase separation affect BaP gas-particle partitioning and heterogeneous degradation (Pye et al., 2017; Schmedding et al., 2020)?

Response:

Thank you for your valuable comment. We appreciate your suggestion to improve the "Conclusion and Discussion" section. In response, we have summarized the conclusion for better clarity. Additionally, we have expanded the discussion to address the potential impact of the phase state of organic coatings on BaP gas-particle partitioning and heterogeneous degradation. Relevant references, such as Shiraiwa et al. (2017), Li et al. (2020), and others, have been incorporated to highlight the recent developments in this area. Please refer to the updated section "Conclusion and Discussion" for further details.

Specific comments:
(1) Some recent modeling work on PAH should be cited and the simulated BaP concentrations are better compared (Wu et al., 2024).
Response:

Thanks for the comment. We have added the following sentence in main text: "Our simulated seasonal BaP concentrations, particularly for the ROI-T approach, align with a previous study using the same emissions and particulate-BaP degradation approach [Wu et al., 2024]." Please refer to lines 328-329.

(2) Line 164 at Page 5, why the authors did not include the gas-phase reaction of BaP with O3 and NO3?
Response:

According to Keyte et al. (2013), most BaP and its oxidized products are observed as particulate phase. Their study also indicates that gas-phase PAHs primarily react with OH, $O_3$, and $NO_3$. Keyte et al. (2013) also summarized the chemical reactivity of PAHs, noting that the second-order reaction rate coefficients for gas-phase reactions of PAHs

with OH radicals are typically around $10^{-11}$ cm$^3$ molec$^{-1}$ s$^{-1}$, while the reaction rate with O$_3$ is six orders of magnitude slower ($10^{-18}$ cm$^3$ molec$^{-1}$ s$^{-1}$). Additionally, no measured value is available for the gas-phase reaction of BaP with O$_3$ in Keyte et al. (2013). Therefore, in this study, we only consider the gas-phase reaction oxidation of BaP by OH.

(3) Line 348-350 at Page 11, the authors wrote that a large portion of the total Bap was still oxidized though the effect of organic coating had been considered in bulk diffusion of BaP in the ROI-T scheme, which was surprising. Again, referring to my major comment 1, the authors should clarify that the effect of viscosity on gas-particle partitioning and BaP degradation rate is related to two different processes. Did the ROI-T scheme consider the phase state effect on bulk diffusivity of BaP in your simulation? I think the result showed in Line 348-350 is not surprising as the reaction rate of BaP with ozone in the particle phase was dependent on RH and T, thus the viscosity of organic shell, which resulted in limited effect of organic coating on BaP degradation in areas with low viscous particles.

Response:

Thank you for your valuable comment.

The ROI-T approach, which models the multiphase degradation of BaP with ozone, incorporates both mass transport and chemical reactions of particle-bound species in the bulk phase and at the surface. The first-order decay rates of BaP, as presented in Table S3 of Mu et al. (2018), are parameterized values that already include the influence of SOA viscosity on bulk diffusivity of BaP.

As mentioned above, we acknowledge that incorporating the viscosity of the organic coatings could provide a more comprehensive representation of gas-particle partitioning, and we have revised the "conclusion and discussion" section to address this comment.

(4) Line 198-200 on Page 6, please double check. I did not see this from Table S3 in Mu et al. (2018).

Response:

Thank you for your comment. The statement "First-order reaction rate coefficients for BaP ozonolysis are sensitive to both temperature and RH below room temperature (296 K), but are only temperature sensitive above room temperature [Mu et al., 2018]" is not from Table S3 in Mu et al. (2018), but from Figure 1A in the same study. As shown in Figure 1A, when the temperature is below 296 K, there is a significant deviation in the first-order multiphase degradation rate coefficient, and its value can differ by 1-2 orders of magnitude under different humidity conditions (temperature < 253 K). In contrast, above 296 K, the degradation rate coefficients remain consistent within the same order

of magnitude. We have added Table S1 for comparing the first-order reaction rate coefficients (k, s$^{-1}$) across the three approaches.

**References**

Chen, Y., Zaveri, R. A., Vandergrift, G. W., Cheng, Z., China, S., Zelenyuk, A., and Shilling, J. E.: Nonequilibrium behavior in isoprene secondary organic aerosol, Environ. Sci. Technol., 57, 38, 14182-14193, https://doi.org/10.1021/acs.est.3c03532, 2023.

Keyte, I. J., Harrison, R. M., and Lammel, G.: Chemical reactivity and long-range transport potential of polycyclic aromatic hydrocarbons--a review, Chem. Soc. Rev., 42(24), 9333-9391, https://doi.org/10.1039/C3CS60147A, 2013.

Li, Y., Day, D. A., Stark, H., Jimenez, J. L., and Shiraiwa, M.: Predictions of the glass transition temperature and viscosity of organic aerosols from volatility distributions, Atmos. Chem. Phys., 20, 8103-8122, 10.5194/acp-20-8103-2020, 2020.

Liu, Y., Shilling, J. E., Schneider, J., Schulz, C., Zöger, M., Martin, S. T., Ye, J., Guenther, A., Souza, R. F., Wendisch, M., and Pöschl, U.: Tight Coupling of Surface and In-Plant Biochemistry and Convection Governs Key Fine Particulate Components over the Amazon Rainforest, ACS Earth and Space Chemistry, 6, 380-390, 10.1021/acsearthspacechem.1c00356, 2022.

Mu, Q., Shiraiwa, M., Octaviani, M., Ma, N., Ding, A., Su, H., Lammel, G., Poschl, U., and Cheng, Y.: Temperature effect on phase state and reactivity controls atmospheric multiphase chemistry and transport of PAHs, Sci. Adv., 4(3), doi:10.1126/sciadv.aap7314, 2018.

Pye, H. O. T., Murphy, B. N., Xu, L., Ng, N. L., Carlton, A. G., Guo, H., Weber, R., Vasilakos, P., Appel, K. W., Budisulistiorini, S. H., Surratt, J. D., Nenes, A., Hu, W., Jimenez, J. L., Isaacman-VanWertz, G., Misztal, P. K., and Goldstein, A. H.: On the implications of aerosol liquid water and phase separation for organic aerosol mass, Atmos. Chem. Phys., 17, 343-369, 10.5194/acp-17-343-2017, 2017.

Rasool, Q. Z., Shrivastava, M., Octaviani, M., Zhao, B., Gaudet, B., and Liu, Y.: Modeling volatility-based aerosol phase state predictions in the Amazon rainforest, ACS Earth and Space Chem., 5, 10, 2910-2924, https://doi.org/10.1021/acsearthspacechem.1c00255, 2021.

Schmedding, R., Rasool, Q. Z., Zhang, Y., Pye, H. O. T., Zhang, H., Chen, Y., Surratt, J. D., Lopez-Hilfiker, F. D., Thornton, J. A., Goldstein, A. H., and Vizuete, W.: Predicting secondary organic aerosol phase state and viscosity and its effect on multiphase chemistry in a regional-scale air quality model, Atmos. Chem. Phys., 20, 8201-8225, 10.5194/acp-20-8201-2020, 2020.

Shiraiwa, M., Li, Y., Tsimpidi, A. P., Karydis, V. A., Berkemeier, T., Pandis, S. N., Lelieveld, J., Koop, T., and Pöschl, U.: Global distribution of particle phase state in atmospheric secondary organic aerosols, Nat. Commun., 8, 15002, 10.1038/ncomms15002, 2017.

Shrivastava, M., Rasool, Q. Z., Zhao, B., Octaviani, M., Zaveri, R. A., Zelenyuk, A., Gaudet, B., Wu, Z., Chen, X., Wang, Z., Chen, H., Wang, Z., Mu, Q., Wu, L., Wang, W., Tang, X., Li, J., Li, Y., Wu, Q., Wang, Y., Zou, Z., and Jiang, Z.: Modeling of polycyclic aromatic hydrocarbons (PAHs) from global to regional scales: model development (IAP-AACM_PAH v1.0) and investigation of health risks in 2013 and 2018 in China, Geosci. Model Dev., 17, 8885-8907, 10.5194/gmd-17-8885-2024, 2024.

Zhang, Y., Nichman, L., Spencer, P., Jung, J. I., Lee, A., Heffernan, B. K., Gold, A., Zhang, Z., Chen,

Y., Canagaratna, M. R., Jayne, J. T., Worsnop, D. R., Onasch, T. B., Surratt, J. D., Chandler, D., Davidovits, P., and Kolb, C. E.: The Cooling Rate and Volatility Dependent Glass Forming Properties of Organic Aerosols Measured by Broadband Dielectric Spectroscopy, Environmental Science & Technology, 10.1021/acs.est.9b03317, 2019.

Zhang, Z., Li, Y., Ran, H., An, J., Qu, Y., Zhou, W., Xu, W., Hu, W., Xie, H., Wang, Z., Sun, Y., and Shiraiwa, M.: Simulated phase state and viscosity of secondary organic aerosols over China, Atmos. Chem. Phys., 24, 4809-4826, 10.5194/acp-24-4809-2024, 2024.

---

## Author Comment (AC2)

Review 1 for "Uncertainties in the effects of organic aerosol coatings on polycyclic aromatic hydrocarbon concentrations and their estimated health effects" by Lou et al.

General comments

This study uses the CAM5 model to investigate the impact of particle-bound PAH degradation approaches on the spatial distribution of BaP and its lung cancer risk, considering the presence or absence of OA coatings. Three representative PAH degradation approaches are implemented. Though PAH modeling and discussion about different degradation schemes are not novel, especially when PAH modeling using the CAM model was already published, it's still fresh to see the simulated oxidized BaP concentrations being compared with measurements. In general, it's well-written and of high scientific quality. This manuscript can be published after addressing the following issues.

Thank you for your valuable comments and suggestions. We have carefully addressed each point as follows.

Specific comments

This study tries to show the uncertainty from different BaP degradation schemes, but what are the criteria for selecting the degradation schemes to quantify the uncertainty range? There are other schemes such as Poschl et al. 2001 which should be even faster, so why is it not chosen? Also, for uncertainty study, it would also be nice to have a no reaction scheme for comparison.

Response:

Thank you for your valuable comment. The primary goal of this study is to identify which chemical schemes best simulate the global concentrations of BaP, assesses the strengths and limitations of current PAH modeling approaches, offering insights into future simulations improvements.

Mu et al. (2018) compared the first-order multiphase degradation rate coefficient k ($s^{-1}$) for various laboratory schemes. It is evident that Pöschl et al. (2001) reports the fastest degradation rate, which is an order of magnitude faster than the thin SOA coating described in Zhou et al. (2013) (the NOA scheme used in this study), approximately 100 times faster than the ROI-T scheme at room temperature, and 3-5 orders of magnitude faster at -20°C. Additionally, Friedman & Selin (2012) incorporated the BaP-$O_3$ reaction from Pöschl et al. (2001) into a global model, but their results showed that simulated BaP concentrations were underestimated by at least a factor of 10.

Since recent laboratory studies [Zelenyuk et al. 2012; Zhou et al. 2012; 2013] have shown that the oxidation kinetics of BaP adsorbed onto particles coated by organics, particularly viscous SOA, are highly sensitive to temperature and humidity. Based on

these findings, we focused our comparison on three OA coating schemes. Two of these schemes have been shown to perform well in comparison with previous schemes at background sites, making them particularly suitable for this study.

To clarify the goal of this study, we have revised the last paragraph of the introduction as follows: "Although simulations of PAHs have significantly improved over the past decade [Sehili and Lammel, 2007; Friedman and Selin, 2012; Shen et al., 2014; Shrivastava et al., 2017; Mu et al., 2018; Wu et al., 2024], particularly in terms of lifetime estimation, understanding of the oxidation chemistry remains a key area of development. The oxidation of particle-bound BaP is highly dependent on the concentrations of oxidants (primarily ozone) and the effectiveness of shielding by viscous organic aerosol (OA) coatings, which are influenced by temperature and relative humidity (RH)." "Since assessments of PAH-induced lung cancer risks often rely on modeled BaP concentrations [Shen et al., 2014; Shrivastava et al., 2017; F Han et al., 2020; 2022; Famiyeh et al., 2021; Li et al., 2022; Lou et al., 2023; Wu et al., 2024], uncertainties in these simulations can have significant implications for estimates of PAH exposure and associated human health risks. This study systematically evaluates the uncertainty in simulated BaP concentrations due to varying chemical mechanisms of BaP oxidation, considering seasonal variations. This paper also assesses the strengths and limitations of current PAH modelling approaches, offering insights into future simulation improvements." Please find in lines 120-133.

Section 2.3 introduces several BaP degradation schemes mostly using text. However, it's not clear for the readers to understand those schemes. In the Supplement, expressions/equations of those schemes can be given. (1) For NOA, give the expression of LH mechanism and k. (2) For the Shielded scheme and the ROI-T scheme, use tables to summarize the different reaction expressions. It would be even better if the intercomparison of reaction rate k could be clearly seen through the summary.

Response:

Thanks for your useful comment. We have added following text and Table R1 in supplement as Text S1 and Table S1.

 "**Text S1. Impact of OA coatings on BaP oxidation approaches**
The oxidation lifetimes of particle-bound BaP, driven by ozone, vary from minutes to several hours. This variation depends on factors such as relative humidity (RH), substrate type, and ozone concentrations. In a previous global model simulation, applying the oxidation kinetics for BaP adsorbed onto soot aerosol particles, as reported by Pöschl et al. (2001), resulted in significant under-prediction of BaP concentrations compared to measurements taken at ground sites. In contrast, Zhou et al. (2012; 2013) measured slower oxidation kinetics for BaP adsorbed on ammonium sulphate particles coated with organics, making their findings more conservative. Furthermore, the oxidation of BaP is influenced by relative humidity (RH), occurring more rapidly under higher RH compared to dry conditions. These findings have significantly advanced

PAH simulations, especially regarding lifetime estimates [Sehili and Lammel, 2007; Friedman et al., 2012; Shen et al., 2014; Shrivastava et al., 2017; Mu et al., 2018]. However, uncertainties remain in modeling the oxidation of particle-bound BaP, as it is sensitive to oxidant concentrations (mainly ozone) and the effectiveness of organic aerosol coatings, which are further influenced by temperature and RH. The main objective of this study is to identify the most accurate chemical schemes for simulating global BaP concentrations, evaluate the strengths and limitations of current PAH modeling approaches, and provide insights for    future simulations improvements. To this end, we focus on comparing three OA coating schemes, with their first-order reaction rate coefficients $k(s^{-1})$ summarized in Table R1.

(1) NOA approach:
The heterogeneous oxidation of particle-bound BaP follows the Langmuir-Hinselwood mechanism. The reaction rate of BaP with $O_3$ proceeds at a rate $k$ ($s^{-1}$) is given by Zhou et al. (2013) as:

$$k = \frac{k_{max}K_{O_3}[O_3]}{1+K_{O_3}[O_3]}$$

where $k_{max}$ is the maximum first-order rate coefficient for BaP loss, $K_{O3}$ is the ozone gas-to-surface partition coefficient, and $[O_3]$ is the gas-phase ozone concentration (molec/cm$^3$). In this study, particle-bound BaP reacts rapidly with oxidants like ozone and OH radicals, and the oxidation kinetics measured by Zhou et al. (2012; 2013) for thin SOA coatings are applied consistently, irrespective of coating thickness or temperature and relative humidity variations.

(2) Shielded approach:
Based on our previous work (Shrivastava et al., 2017), the SOA Shielded approach is used to account for the protection of BaP by viscous SOA coatings. When BaP is coated by organic aerosols, the kinetics of its heterogeneous oxidation with ozone slow down significantly. This is because thick OA coatings hinder the mass transfer of BaP from the particle interior to the surface and BaP reacts at the particle surface with ozone. The effectiveness of this shielding depends on the thickness and viscosity of the SOA, which are influenced by temperature and relative humidity [Zhou et al., 2012; Zhou et al., 2013]. In this approach, when SOA coatings are less than 20 nm, they are assumed to be ineffective in shielding particle-bound BaP. As a result, the heterogeneous oxidation kinetics remain unchanged, similar to the NOA approach (for thin SOA coatings in Table R1). For thicker SOA coatings (> 20 nm), the heterogeneous oxidation of particle-bound BaP is essentially inhibited under dry or cool conditions (RH < 50% or temperature < 296 K). The oxidation kinetics of BaP with thick SOA coatings under varying humidity and temperature are measured by Zhou et al. (2012; 2013).

(3) ROI-T approach:
In accordance with Mu et al. (2018), the ROI-T approach is employed, which considers the temperature and humidity dependence of the phase state, diffusivity, and reactivity of particulate-phase BaP. The first-order reaction rate coefficients for BaP ozonolysis

are highly sensitive to both temperature and RH at temperatures below room temperature (296 K), but they become primarily temperature-sensitive above this threshold [Mu et al., 2018]. Under cool and dry conditions, the first-order reaction rate coefficients are three orders of magnitude lower than those under warmer conditions (Table R1). Importantly, the ROI-T approach results in a much slower oxidation of particle-bound BaP compared to the NOA approach under cool, dry conditions but predicts a faster reaction rate under warmer conditions (e.g., 303K). ″

Table R1. Comparison of first-order reaction rate coefficients (k, s$^{-1}$) across the three approaches, calculated at 50 ppb O$_3$. Note: The thickness of SOA coatings in the Shielded scheme must always be considered when interpreting the results.

| Temperature | NOA | Shielded Thin SOA | Thick SOA | ROI-T |
|---|---|---|---|---|
| RH=70% | | | | |
| 303$K$ | $2.6\times10^{-4}$ | $2.6\times10^{-4}$ | $2.9\times10^{-4}$ | $5.8\times10^{-4}$ |
| 288K | $2.7\times10^{-4}$ | $2.7\times10^{-4}$ | 0 | $1.6\times10^{-4}$ |
| 273K | $2.9\times10^{-4}$ | $2.9\times10^{-4}$ | 0 | $4.0\times10^{-5}$ |
| 263K | $3.0\times10^{-4}$ | $3.0\times10^{-4}$ | 0 | $1.6\times10^{-5}$ |
| 253K | $3.1\times10^{-4}$ | $3.1\times10^{-4}$ | 0 | $5.6\times10^{-6}$ |
| RH=50% | | | | |
| 303$K$ | $2.6\times10^{-4}$ | $2.6\times10^{-4}$ | 0 | $5.7\times10^{-4}$ |
| 288K | $2.7\times10^{-4}$ | $2.7\times10^{-4}$ | 0 | $1.5\times10^{-4}$ |
| 273K | $2.9\times10^{-4}$ | $2.9\times10^{-4}$ | 0 | $2.2\times10^{-5}$ |
| 263K | $3.0\times10^{-4}$ | $3.0\times10^{-4}$ | 0 | $2.0\times10^{-6}$ |
| 253K | $3.1\times10^{-4}$ | $3.1\times10^{-4}$ | 0 | $1.8\times10^{-7}$ |
| RH=30% | | | | |
| 303$K$ | $2.6\times10^{-4}$ | $2.6\times10^{-4}$ | 0 | $5.3\times10^{-4}$ |
| 288K | $2.7\times10^{-4}$ | $2.7\times10^{-4}$ | 0 | $1.0\times10^{-4}$ |
| 273K | $2.9\times10^{-4}$ | $2.9\times10^{-4}$ | 0 | $4.3\times10^{-6}$ |
| 263K | $3.0\times10^{-4}$ | $3.0\times10^{-4}$ | 0 | $4.9\times10^{-7}$ |
| 253K | $3.1\times10^{-4}$ | $3.1\times10^{-4}$ | 0 | $1.5\times10^{-7}$ |

L78-79 It seems that ppLFER is good for anthropogenically impacted areas but Junge-Pankow is good for remote areas. So why do you choose ppLFER in the end? Have you tested their differences in your model?

Response:

Thank you for your insightful comment. Pankow (1994) developed the modeling framework for gas-particle partitioning of semivolatile organic compounds (SOCs) in the atmosphere. The partitioning coefficient, $K_p$, is expressed as:

$$K_p = \frac{C_p}{C_g TSP} = \frac{N_s a_{TSP} T e^{(Q_1 - Q_v)/RT}}{1600 p_L^0} \qquad (R1)$$

where $C_p$ and $C_g$ represent the particle and gas-phase concentrations (ng m$^{-3}$), $N_s$ is the surface concentration of sorption sites (sites cm$^{-2}$), $a_{TSP}$ is the specific surface area of total suspended particles (TSP) (m$^2$ g$^{-1}$), $Q_1$ and $Q_v$ are the enthalpies of desorption and vaporization (kJ mol$^{-1}$), R is the gas constant, T is the temperature (K), and $p_L^0$ is the vapor pressure of the compound as a subcooled liquid.

SOCs in the atmosphere tend to partition through absorption into the organic matter (OM), which includes both primary organic aerosol and secondary organic aerosol, rather than being adsorbed on the particle surface. Pankow (1994) demonstrated that the use of the vapor pressure $p_L^0$ as a descriptor for partitioning was valid for both adsorption and absorption.

Subsequent developments extended the model's applicability to laboratory measurements. Pankow (1994) replaced $p_L^0$ with the octanol-air partitioning coefficient ($K_{oa}$), leading to the following formulation:

$$K_p = f_{OM} \frac{MW_{oct} \gamma_{oct}}{MW_{OM} \gamma_{OM} 10^{12} \rho_{OM}} K_{oa} \qquad (R2)$$

where $f_{OM}$ is the density of OM, $\gamma$ are activity coefficients of the SOC in both OM and octanol, and the partitioning coefficients are derived from laboratory studies (Dachs and Eisenreich 2000; Lohmann and Lammel, 2004).

In our previous work, we also explored the use of single-parameter linear free energy relationships (sp-LFER) to model the dual absorption-adsorption behavior in OA and black carbon (BC), an approach derived from Junge-Pankow's framework. The recently-developed pp-LFER approach, however, distinguishes itself by treating OA as consisting of two distinct phases: a liquid phase (comprising both water-soluble and organic-soluble components) and a semi-solid/solid organic polymer phase. Shahpoury et al. (2016) describe the overall partitioning coefficient $K_p$ as:

$$K_p = 10^{-12} \left[ \left( K_{BC-air} \times \frac{BC}{TSP} \times a_{BC} \times 10^6 \right) + \left( \frac{K_{DMSO-air}}{\rho_{DMSO}} \times \frac{OA_A}{TSP} \right) + \left( K_{PU-air} \times \frac{OA_B}{TSP} \right) \right]$$

Here, the first term represents adsorption of PAH on BC, with $K_{BC-air}$ as the soot-air partitioning coefficient, $a_{BC}$ as the specific surface area of soot, and BC as soot concentration. The second and third terms represent absorptive partitioning of PAH within two distinct OA phases, $OA_A$ and $OA_B$, with the respective partitioning coefficients for dimethyl sulfoxide (DMSO) and polyurethane ether (PU). The temperature dependency of $K_p$ is incorporated using the van't Hoff relationship. The details for calculating the partitioning coefficients $K_{BC-air}$, $K_{DMSO-air}$ and $K_{PU-air}$ in this study are provided in our previous study (Shrivastava et al., 2017).

It is important to note that the pp-LFER model predicts more than 90% of particle-bound BaP is absorbed by OA, while previous regional and global models suggest that most PAH is bound to BC (Friedman and Selin, 2012; Shen et al., 2014). This discrepancy is largely due to significant differences in partitioning coefficients used for BC in these models. Our previous work demonstrated that the $K_{BC\text{-}air}$ value derived from Lohmann and Lammel (2004) is two orders of magnitude higher than the pp-LFER-derived value at 298 K. This suggests that previous models may have overestimated the contribution of BC to particle-bound PAHs. Furthermore, using pp-LFER results in a moderate (~5-fold) increase at 298 K in the effective partitioning coefficient of BaP to OA compared to approaches such as Odabasi et al. (2006), which used n-octanol as a surrogate for OA. However, n-octanol is not an ideal surrogate for atmospheric OA due to its low water solubility and low polarity, making the pp-LFER approach, with its two-phase OA model, a more accurate representation of atmospheric SOA (Shahpoury et al., 2016).

Therefore, the choice of the pp-LFER approach over the Junge-Pankow model is driven by its more realistic representation of the complex partitioning behavior in the atmosphere, particularly the dual-phase nature of organic aerosol. For a global model, a unified approach is necessary to account for both remote and urban areas. While the Junge-Pankow model is effective for remote regions, the pp-LFER model offers a more accurate representation of urban and anthropogenically impacted areas, where both adsorption and absorption processes play significant roles.

We have added text S2 in supplementary.

When discussing the role of emission and meteorology in influencing BaP level and distribution, cited references are used. However, those references use different emission inventories and meteorological inputs, so results from those references can't be directly used. It would be more straightforward to use this model's inputs for analysis.

Response:

Thank you for your comment, and apologies for any confusion. The references you are referring to, namely Shrivastava et al. (2017) and Lou et al. (2023), are based on our previous studies, and both used the same emission inventories and meteorological inputs. To clarify this in the manuscript, we have updated the text "Our previous studies, using the same simulations, reported that while the coarse-grid model significantly underestimates concentrations in urban-affected regions, the downscaled BaP vastly improves the comparison between the model and observations [Shrivastava et al., 2017; Lou et al., 2023]." Please refer to lines 433-435.

Technical comments

L33 Explain the abbreviation of OA.

Response:

We provide the full name for all abbreviations upon their first use. In the case of "OA", it is first introduced as "organic aerosol (OA)" in the Abstract: "Three degradation approaches, each reflecting varying effects of organic aerosol (OA) coatings on BaP degradation are included in this study." Please refer to lines 32-34.

L35 Not only related to temperature but also relative humidity.

Response:

Thank you for your valuable suggestion. We have revised the sentence for clarity as follows: "ROI-T (viscous OA coatings slowing down PAHs oxidization in response to temperature and relative humidity (RH))". Please refer to lines 35-36.

L36 The influence missed meteorology (transport and other meteo conditions) which you also explained later.

Response:

Thank you for pointing it out. We have revised the sentence for clarity as follows: "Our findings reveal that the seasonal variation of BaP is highly dependent on changes in emissions, deposition, transport, and the chosen degradation approach, all of which are influenced by meteorological conditions". Please refer to lines 36-38.

L84 Explain the abbreviation of OH, NO3.

Response:

Thanks for your suggestion. We have revised the sentence as follows: "As a semi-volatile compound, BaP in the gas-phase undergoes degradation through various pathways, primarily involving reactions with hydroxyl radicals (OH) and nitrate radicals ($NO_3$)…". Please refer to lines 91-93.

L122 Lung cancer risk part is not mentioned in section 3, which is important.

Response:

Thanks for your important suggestion. We have revised the sentence as follows: "Section 3 first presents the simulated fresh and oxidized BaP concentrations in winter and summer, followed by a detailed comparison between simulated BaP and measurements, as well as an assessment of PAH-related lung cancer risk". Please refer to lines 135-137.

L135 suburban background site.

Response:

Revised. Thanks!

L158 Delete "organic aerosols" as OA is already explained earlier.

Response:

Revised. Thanks!

L164 Delete "hydroxyl radicals" as OH should be explained earlier.

Response:

Revised. Thanks!

L190 How does the model decide if the coating is thicker than 20 nm?

Response:

We calculate the organic aerosol (OA) coating thickness around a black carbon (BC) core for each model grid and time step. OA coating thickness evolves due to atmospheric processes like the condensation of organic vapors on preexisting aerosols including the formation of secondary organic aerosols (SOA). Previous research has indicated that the threshold for effective OA coating to shield BaP from oxidation ranges between 10 and 20 nm [Zhou et al., 2012; 2013; Keyte et al., 2013]. Therefore,

we adopt a conservative coating-thickness threshold of 20 nm, where coatings thicker than this are assumed to provide effective shielding, particularly under cool and dry conditions. Coatings thinner than 20 nm are treated as insufficient to shield, allowing BaP to react with $O_3$. For further details, refer to Shrivastava et al. (2017).

L194-195 How does temperature influence the reaction rates? I only see that RH plays a role.

Response:

Sorry for the confusing description. We have revised the sentence to clarify the influence of both temperature and RH on the heterogeneous loss kinetics of particle-bound BaP for Shielded approach. The role of temperature is significant, as it works in conjunction with RH to determine the oxidation kinetics of BaP. Specifically, under dry or cool conditions (RH < 50% or temperature < 296 K), the thick SOA coatings (> 20 nm) completely inhibit the heterogeneous loss of BaP. In contrast, under humid (RH > 50%) and warm (temperature > 296 K) conditions, different oxidation kinetics occur. We have revised the sentence as: "Different oxidation kinetics with ozone are applied under humid (RH $\geq$ 50%) and warm (temperature $\geq$ 296 K) conditions with thick SOA coatings..." Please refer to lines 216-218.

L210 The explanation of the Shielded scheme is not complete. Since it's not easy to explain it in one sentence in the brackets, it's better not to explain it at all. Section 2.3 has explained it well.

Response:

Thank you for your suggestion. We have removed all the brief explanations in section 2.4.

L214 Which are the two simulation years?

Response:

Sorry for the confusing description. We have revised the sentence as follows: "All simulations are conducted over two years (2007-2008), with the first year allocated for spin-up." Please refer to line 247.

L226 GFED4 is already widely used by all kinds of models replacing GFED3. So why is GFED3 still used in the simulation? Does it influence the results?

Response:

Thank you for your comment. Our study series began in 2016, at which time GFED3 was commonly used, and we have maintained consistency with that dataset throughout our analyses.

In this study, only open biomass burning OC emissions are obtained from the GFED3, meaning that its impact is limited to the formation of OA coatings. In response to the review's comment, we compared the OC emissions from GFED3 and GFED4 for the year 2008. The global OC budget from GFED4 is 13.8 Tg (available from https://www.geo.vu.nl/~gwerf/GFED/GFED4/tables/GFED4.1s_OC.txt), while from GFED3 it is 15.6 Tg - approximately13% higher. This suggests that our use of GFED3 results in slightly thicker OA coatings, which could, in principle, influence the particle-bound BaP.

However, BaP concentrations are predominantly high in wintertime in East Asia and South Asia, and urban areas in Africa and Europe, where anthropogenic emissions from industrial and residential sources contribute substantial amounts of organic aerosols. As a result, BaP degradation in these regions is not particularly sensitive to OA coatings from open fires. In addition, wildfires typically occur in remote areas, limiting their direct impact on PAH-related human health.

Therefore, the choice between GFED3 and GFED4 does not significantly affect the overall results of our study.

L238 Does n equal to 8? If yes, just use 8.

Response:

Thank you for your comment. n is not equal to 8; instead, it represents the total number of contributing emission grid cells (0.1°×0.1°) within the nine-grid-cell neighborhoods. For a receiving grid, n is approximately 4275 (with about 475 emission grid cells within each 2.5°×1.9° grid cell), though the actual value depends on the alignment of the grid resolutions. We have revised section 2.6 in the main text to clarify this. Please refer to lines 268-273, and 280-281.

L240 What does "1 to 16" mean?

Response:

The phrase "1 to 16" refers to the 16 wind directions considered in the analysis. These directions are: N, NNE, NE, NEE, E, SEE, SE, SSE, S, SSW, SW, NWW, NW, and NNW. We have revised section 2.6 in the main text to clarify this. Please refer to lines 276-277.

Section 2.7 What are the values of CSF and SUS or how are they calculated? For LADD calculation, what's the value of inhalation rate, exposure duration and body weight?

Response:

Thanks for your comment. The Incremental Lifetime Cancer Risk is calculated following Shen et al. (2014). According to their study, a cancer slop factor (CSF) of 26.6 kg (body weight)·day/mg for BaP was adopted as the maximum likelihood estimate based on epidemiological data from studies on coke oven workers, using a multistage type model [U.S. EPA, 1982]. The uncertainty of the CSF was determined based on the variability observed across different epidemiological studies [Bostrom et al., 2002], with a standard deviation of 0.38 for the log-transformed CSF [Shen et al., 2014].

Overall susceptibility (SUS) accounts for individual variations in susceptibility and is defined as the product of genetic susceptibility factor (GeneSus), ethnicity-adjusted factor (EAF), and age-sensitivity factor (ASF), respectively. GeneSus represents the impact of genetic variations on an individual's susceptibility to BaP-induced cancer risk. Different genotypes may lead to variations in metabolic activation or detoxification of PAHs, affecting carcinogenic risk. EAF was calculated based on the lung cancer incidences for individual ethnicities reported by the United States Cancer Statistics (available from https://www.cdc.gov/united-states-cancer-statistics/index.html), excluding the effects of smoking. ASF accounts for age-related differences in susceptibility to BaP exposure. Weighting cancer risk by a factor of 10, 2, and 1 were used for the age groups of < 2, 2-16, and > 16 years, respectively [CA EPA, 2009].

We have added revised section 2.7 for clarify. Please refer to lines 294-305.

L271 How is PWGA calculated?

Response:

The population weighted global average BaP is calculated as follows: After downscaling and obtaining a finer horizontal resolution of BaP concentrations, the

population density corresponding to each 0.1°×0.1° grid is used as a weight factor. The BaP concentrations for all grid are then summed, with each value weighted according to its respective population density, and the total is averaged based on the global population distribution. The formula is as follows:

$$PWGA = \frac{\sum C_i \times P_i}{\sum P_i} \quad (F1)$$

Where $C_i$ and $P_i$ are BaP concentration and population density for grid i, respectively.

L276 Use PWGA as it's explained earlier.

Response:

Revised. Thanks.

L280 Should be "lighting" not "lightning".

Response:

Revised. Thanks.

L279 Shen et al. 2013 used a different BaP emission inventory. Instead of citing the emission results from the literature, it's better to calculate them from the model's emission inventory, which is more convincing.

Response:

Sorry for the confusing description. The BaP emission inventory used in this study is the Global Emissions Modeling System (GEMS) with a 0.1º × 0.1º resolution (gems.sustech.edu.cn). This inventory was derived from the PKU-PAH Global Emission Inventory, which was originally developed by Shen et al. (2013). The PKU-PAH inventory includes emissions from 69 major fuel consumption sources and accounts for an uncertainty range of -40% to +60% [Shen et al., 2013]. For 2008, BaP emissions were allocated to various sectors such as residential biofuel, residential fossil fuel, industry, transportation, agricultural waste burning, and open-fire biomass burning. These allocations were made using regional ratios provided in Shen et al. (2013), with residential biomass use representing over 60% of the global BaP emissions. We clarify this in lines 252-254.

Figure 1 Unit is missing in the color bar.

Response:

Revised. Thanks.

L301 Does OH really have the same trait as ozone here?

Response:

In section 2.3, we state: "Heterogeneous reactions of particulate-phase BaP with OH and ozone are included in the model [Cazaunau et al., 2010; Zhou et al., 2012; 2013; Keyte et al., 2013]. The second-order rate coefficient for the reaction of particle-bound BaP with OH is determined to be $2.9\times10^{-13}$ $cm^3$ molecules$^{-1}$ s$^{-1}$ [Esteve et al., 2006], which is two orders of magnitude slower than the gas-phase reaction rate of BaP with OH. Conversely, particle-bound BaP reacts rapidly with ozone within a few hours, representing the primary oxidation pathway for BaP." Therefore, OH oxidation is less important compared to $O_3$ oxidation. However, the viscosity of OA coatings can influence all particle-bound BaP oxidation processes.

L308 Temperature, humidity and emissions can be plotted, instead of citing references as references may not use the same inputs.

Response:

Thanks for your suggestion. We have revised Fig. 2 (Figure R1) as follows, including temperature, humidity, and emissions.

[Figure]

**Figure R1. The spatial distribution of surface-layer average temperature (top panel, unit: K), relative humidity (middle panel, unit: %), and spatial distribution of BaP emissions (bottom panel, unit kg/grid cell) in DJF (December-January-February) and JJA (June-July-August), respectively.**

L337 Can you show the chemical reaction and reaction rate for producing oxidized-BaP?

Response:

After calculating the heterogeneous reaction rate $k_1$ for each time step, we assume that fresh particulate-phase BaP reacts with ozone at this rate to form oxidized BaP. A comparison of the $k_1$ from the three approaches is presented in Table R1 (now is Table S1).

L389 Since NMB for the background sites is -18%, could downgrading the background sites improve model performance?

Response:

We compared observed surface BaP concentrations with downscaled simulated BaP concentrations as follows. In this case, the simulated BaP concentrations at fine resolution tend to overestimate the measurements, as most observation sites are in

human-impacted area. Conversely, the coarse-resolution simulations significantly underestimate BaP concentrations.

[Figure]

**Figure R2. Comparison between simulated (after downscaling) surface BaP concentrations from NOA, Shielded, and ROI-T simulations and ground-based measurements for 66 background sites. The circle area is proportional to the number of days sampled at each site.**

Among the various simulation approaches, the NOA approach shows the best agreement with measurements, with a normalized mean bias (NMB) within +10% in both DJF and JJA (Fig. R2). The ROI-T approach has an NMB within +30% in both seasons, while the Shielded approach overestimates fresh BaP by approximately 160% (Fig. R2). Although the NMB values are better for the NOA and ROI-T approaches with downscaled concentrations compared to the coarse-resolution results, the low NMB values at fine resolution reflect substantial deviations in BaP concentrations (i.e., both overestimates and underestimates) across different sites (see Fig. R2 and Fig. 5 in main text).

Additionally, most of the background sites represent larger areas (e.g., 100 km×100 km), unlike the smaller 10×10 km grid used for the downscaled simulations. Therefore, comparing observed BaP concentrations at background sites is more appropriate with simulations at the original grid resolution (~200 km×200 km).

Figure 5. Why are China and Europe chosen to be shown? No explanation in the text.

Response:

Thanks for the comment. The regions of China and Europe were selected based on the following criteria: (1) each region has at least 10 measurement sites, and (2) the data from these sites span more than one year, allowing for a representation of seasonal variations. Both China and Europe meet these criteria, with 18 sites each. We have revised the main text to address this comment. Please refer to lines 475-477.

L419 Use NMB.

Response:

Revised. Thanks!

L427 Calculate the emission instead of citing a reference that uses another inventory.

Response:

As mentioned above. The BaP emission inventory used in this study is the Global Emissions Modeling System (GEMS) with a 0.1º × 0.1º resolution (gems.sustech.edu.cn). This inventory was derived from the PKU-PAH Global Emission Inventory, which was originally developed by Shen et al. (2013).

L455-457 Site description should be moved to site introduction part.

Response:

Thank you for the suggestion. We have revised the manuscript. Please see details in lines 152-158.

Figure 7: Model results may also have error bars as observations?

Response:

The observed data for the Grenoble site covers the entire year of 2013, while for the SIRTA site, data spans from November 2014 to December 2015. These observed data also have a detection limit of 0.001 ng. In contrast, the simulated BaP values correspond to 2008. Unlike the observed data, the simulated BaP values do not have a lower detection limit, so the 15th percentile of the simulated BaP is often much lower than the observed values. Additionally, differences in emissions and meteorological conditions between 2008 and 2014 should introduce a bias between the simulated and observed BaP concentrations.

The error bars in Figure 7 represent the range of observed values from the 15th to the 85th percentile, while the simulation results only show the median value to assess the model's ability to capture seasonal variation and the observed magnitude. Therefore, including error bars for the model results would not enhance the figure or affect the study's conclusion.

L478 Why "however"?

Response:

"however" is really not appropriate here and we have removed it. Thanks!

L508 "selected".

Response:

Revised. Thanks!

L590 "effect"

Response:

We have revised "Conclusion and Discussion". Thanks!

**References:**

Bostrom, C. E., Gerde, P. , Hanberg, A., Jernstrom, B., Johansson, C., Kyrklund, T., Rannug, A., Tornqvist, M., Victorin, K., and Westerholm, R.: Cancer risk assessment, indicators, and guidelines for polycyclic aromatic hydrocarbons in the ambient air, Environ. Health Perspect., 110, 451-488, doi:10.1289/ehp.110-1241197, 2002.

CA EPA. Technical support document for cancer potency factors: methodologies for derivation, listing of available values, and adjustments to allow for early life stage exposures https://oehha.ca.gov/media/downloads/crnr/tsdcancerpotency.pdf (accessed 4 Feb. 2025)

Cazaunau, M., Ménach, K. Le, Budzinski, H., and Villenave, E.: Atmospheric heterogeneous reactions of Benzo(a)pyrene, Z. für Phys. Chem., 224, 1151-1170, https://doi.org/10.1524/zpch.2010.6145, 2010.

Esteve, W., Budzinski, H., and Villenave, E.: Relative rate constants for the heterogeneous reactions of NO and OH radicals with polycyclic aromatic hydrocarbons adsorbed on carbonaceous particles. Part 2: PAHs adsorbed on diesel particulate exhaust SRM 1650a, Atmos. Environ., 40(2), 201-211, https://doi.org/10.1016/j.atmosenv.2005.07.053, 2006.

Famiyeh, L., Chen, K., Xu, J., Sun, Y., Guo, Q., Wang, C., Lv, J., Tang, Y.-T., Yu, H., Snape, C., He, J.: A review on analysis methods, source identification, and cancer risk evaluation of atmospheric polycyclic aromatic hydrocarbons, Sci. Total. Environ., 789, 147741, 2021.

Dachs, J., and Eisenreich, S. J.: Adsorption onto aerosol soot carbon dominates gas-particle partitioning of polycyclic aromatic hydrocarbons, Environ. Sci. Technol., 34(17), 3690-3697, https://doi.org/10.1021/es991201+, 2000.

Friedman, C. L. and Selin, N. E.: Long-range atmospheric transport of polycyclic aromatic hydrocarbons: A global 3-D model analysis including evaluation of Arctic sources, Environ. Sci. Technol., 46(17), 9501-9510, https://doi.org/10.1021/es301904d, 2012.

Han, F., Guo, H., Hu, J., Zhang, J., Ying, Q., and Zhang, H.: Sources and health risks of ambient polycyclic aromatic hydrocarbons in China, Sci Total Environ, 698, 134229, https://doi.org/10.1016/j.scitoteenv.2019.134229, 2020.

Han, F., Kota, S. H., Sharma, S., Zhang, J., Ying, Q., and Zhang, H.: Modeling polycyclic aromatic hydrocarbons in India: Seasonal variations, sources and associated health risks, Environ Res, 212(Pt D), 113466, https://doi.org/10.1016/j.envres.2022.113466, 2022.

Keyte, I. J., Harrison, R. M., and Lammel, G.: Chemical reactivity and long-range transport potential of polycyclic aromatic hydrocarbons--a review, Chem. Soc. Rev., 42(24), 9333-9391, https://doi.org/10.1039/C3CS60147A, 2013.

Li, Y., Zhu, Y., Liu, W., Yu, S., Tao, S., and Liu, W.: Modeling multimedia fate and health risk assessment of polycyclic aromatic hydrocarbons (PAHs) in the coastal regions of the Bohai and Yellow Seas, Sci. Total Environ., 818, 151789, https://doi.org/10.1016/j.scitotenv.2021.151789, 2022.

Lohmann, R., and Lammel, G.: Adsorptive and absorptive contributions to the gas-particle partitioning of polycyclic aromatic hydrocarbons: state of knowledge and recommended parametrization for modeling, Environ. Sci. Technol., 38(14), 3793-3803, https://doi.org/10.1021/es035337q, 2004.

Lou, S., Shrivastava, M., Ding, A., Easter, R. C., Fast, J. D., Rasch, P. J., Shen, H., Massey Simonich, S. L., Smith, S. J., and Tao, S.: Shift in Peaks of PAH‐Associated Health Risks From East Asia to South Asia and Africa in the Future, Earth's Future, 11(6), e2022EF003185, https://doi.org/10.1029/2022EF003185, 2023.

Mu, Q., Shiraiwa, M., Octaviani, M., Ma, N., Ding, A., Su, H., Lammel, G., Poschl, U., and Cheng, Y.: Temperature effect on phase state and reactivity controls atmospheric multiphase chemistry and transport of PAHs, Sci. Adv., 4(3), doi:10.1126/sciadv.aap7314, 2018.

Odabasi, M., Cetin, E., Sofuoglu, A.: Determination of octanol-air partition coefficients and supercooled liquid vapor pressures of PAHs as a function of temperature: Application to gas-particle partitioning in an urban atmosphere, Atmos. Environ., 40(34), 6615-6625, https://doi.org/10.1016/j.atmosenv.2006.05.051, 2006.

Pankow, J. F.: An absorption model of gas/particle partitioning of organic compounds in the atmosphere, Atmos. Environ., 48(2), 185-188, doi:10.1016/1352-2310(94)90093-0.

Pöschl, U., Letzel, T., Schauer, C., and Niessner, R.: Interaction of ozone and water vapor with spark discharge soot aerosol particles coated with benzo [a] pyrene: O3 and H2O adsorption, benzo [a] pyrene degradation, and atmospheric implications, J. Phys. Chem. A, 105(16), 4029-4041, https://doi.org/10.1021/jp004137n, 2001.

Sehili, A. M., and Lammel, G.: Global fate and distribution of polycyclic aromatic hydrocarbons emitted from Europe and Russia, Atmos. Environ., 41(37), 8301-8315, https://doi.org/10.1016/j.atmosenv.2007.06.050, 2007.

Shahpoury, P., Lammel, G., Albinet, A., Sofuoglu, A., Dumanoglu, Y., Sofuoglu, S. C., Wagner, Z., and Zdimal, V.: Evaluation of a Conceptual Model for Gas-Particle Partitioning of Polycyclic

Aromatic Hydrocarbons Using Polyparameter Linear Free Energy Relationships, Environ. Sci. Technol., 50(22), 12312-12319, https://doi.org/10.1021/acs.est.6b02158, 2016.

Shen, H., Huang, Y., Wang, R., Zhu, D., Li, W., Shen, G., Wang, B., Zhang, Y., Chen, Y., Lu, Y., Chen, H., Li, T., Sun, K., Li, B., Liu, W., Liu, J., and Tao, S.: Global Atmospheric Emissions of Polycyclic Aromatic Hydrocarbons from 1960 to 2008 and Future Predictions, Environ. Sci. Technol., 47(12), 6415-6424, doi:10.1021/es400857z, 2013.

Shen, H., Tao, S., Liu, J., Huang, Y., Chen, H., Li, W., Zhang, Y., Chen, Y., Su, S., Lin, N., Xu, Y., Li, B., Wang, X., and Liu, W.: Global lung cancer risk from PAH exposure highly depends on emission sources and individual susceptibility, Sci. Rep., 4, 6561, https://doi.org/10.1038/srep06561, 2014.

Shrivastava, M., Lou, S., Zelenyuk, A., Easter, R. C., Corley, R. A., Thrall, B. D., Rasch, P. J., Fast, J. D., Massey Simonich, S. L., Shen, H., and Tao, S.: Global long-range transport and lung cancer risk from polycyclic aromatic hydrocarbons shielded by coatings of organic aerosol, Proc. Natl. Acad. Sci. U.S.A., 114(6), 1246-1251, https://doi.org/10.1073/pnas.1618475114, 2017.

U.S. EPA. Carcinogen Assessment Of Coke Oven Emissions. U.S. Environmental Protection Agency, Washington, D.C., EPA/600/6-82/003F (NTIS PB84170182), 1982.

Wu, Z., Chen, X., Wang, Z., Chen, H., Wang, Z., Mu, Q., Wu, L., Wang, W., Tang, X., Li, J., Li, Y., Wu, Q., Wang, Y., Zou, Z., and Jiang, Z.: Modeling of polycyclic aromatic hydrocarbons (PAHs) from global to regional scales: model development (IAP-AACM_PAH v1.0) and investigation of health risks in 2013 and 2018 in China, Geosci. Model Dev., 17, 8885-8907, 10.5194/gmd-17-8885-2024, 2024.

Zelenyuk, A., Imre, D., Beranek, J., Abramson, E., Wilson, J., and Shrivastava, M.: Synergy between secondary organic aerosols and long-range transport of polycyclic aromatic hydrocarbons, Environ. Sci. Technol., 46(22), 12459-12466, https://doi.org/10.1021/es302743z, 2012.

Zhou, S., Lee, A. K. Y., McWhinney, R. D., and Abbatt, J. P. D.: Burial Effects of Organic Coatings on the Heterogeneous Reactivity of Particle-Borne Benzo a pyrene (BaP) toward Ozone, J. Phys. Chem. A, 116(26), 7050-7056, https://doi.org/10.1021/jp3030705, 2012.

Zhou, S., Shiraiwa, M., McWhinney, R. D., Poschl, U., and Abbatt, J. P.: Kinetic limitations in gas-particle reactions arising from slow diffusion in secondary organic aerosol, Faraday Discuss, 165, 391-406, https://doi.org/10.1039/C3FD00030C, 2013.